

# Evaluation of Northern Hemisphere snow water equivalent in CMIP6 models with satellite-based SnowCCI data during 1982-2014

Kerttu Kouki[1], Petri Räisänen[1], Kari Luojus[1], Anna Luomaranta[1], Aku Riihelä[1]

[1]Finnish Meteorological Institute, Helsinki, P.O. Box 503, 00101, Finland

*Correspondence to*: Kerttu Kouki (kerttu.kouki@fmi.fi)

**Abstract.** Seasonal snow cover of the Northern Hemisphere (NH) is a major factor in the global climate system, which makes snow cover an important variable in climate models. Monitoring snow water equivalent (SWE) at continental scale is only possible from satellites, yet substantial uncertainties have been reported in NH SWE estimates. A recent bias-correction method significantly reduces the uncertainty of NH SWE estimation, which enables a more reliable analysis of the climate models'

ability to describe the snow cover. We have intercompared the CMIP6 (Coupled Model Intercomparison Project Phase 6) and satellite-based NH SWE estimates north of 40° N for the period 1982-2014, and analyzed with a regression approach whether temperature (T) and precipitation (P) could explain the differences in SWE. We analyzed separately SWE in winter and SWE change rate in spring. The SnowCCI SWE data are based on satellite passive microwave radiometer data and in situ data. The analysis shows that CMIP6 models tend to overestimate SWE, however, large variability exists between models. In winter, P

is the dominant factor causing SWE discrepancies especially in the northern and coastal regions. This is in line with the expectation that even too cold temperatures cannot cause too high SWE without precipitation. T contributes to SWE biases mainly in regions, where T is close to 0°C in winter. In spring, the importance of T in explaining the snowmelt rate discrepancies increases. This is to be expected, because the increase in T is the main factor that causes snow to melt as spring progresses. Furthermore, it is obvious from the results that biases in T or P can not explain all model biases either in SWE in

winter or in the snowmelt rate in spring. Other factors, such as deficiencies in model parameterizations and possibly biases in the observational datasets, also contribute to SWE discrepancies. In particular, linear regression suggests that when the biases in T and P are eliminated, the models generally overestimate the snowmelt rate in spring.

## 1 Introduction

Seasonal snow cover of the Northern Hemisphere (NH) is an important factor of the global climate system. The seasonal snow

cover greatly influences surface albedo and, thus, the Earth's energy balance (Callaghan et al., 2011; Flanner et al., 2011; Qu and Hall, 2005; Trenberth and Fasullo, 2009). This makes snow cover an important variable in climate models (Derksen and Brown, 2012; Loth et al., 1993). Additionally, snow cover significantly affects the hydrological cycle at high latitudes and in mountainous regions (Barnett et al., 2005; Bormann et al., 2018; Callaghan et al., 2011; Douville et al., 2002). In winter, snow cover stores large amounts of fresh water, which limits water availability. In spring and summer, warming temperatures melt

the snowpack, releasing water as runoff. In some areas, snow is the largest freshwater storage, and about one sixth of the
world's population is dependent on meltwater from snow (Barnett et al., 2005; Hall et al., 2008).

Melting snow is also a major source for hydropower (Callaghan et al., 2011; Magnusson et al., 2020). Due to the global
warming, the melting season begins earlier, with the timing of streamflow peaks also becoming earlier (Kundzewicz et al.,
2008). In addition, changes in snow cover affect the intensity of spring streamflow, as an increasing proportion of winter
precipitation is rain instead of snow (Callaghan et al., 2011; Cohen et al., 2015; Dong et al., 2020; Kundzewicz et al., 2008).
Thus, changes in snow cover affecting the hydrological cycle can cause regional water shortages and affect hydropower
production.

SWE (snow water equivalent) is the amount of water contained in the snowpack (in units of kg m$^{-2}$), or equivalently, the height
of the water layer (in units of mm) that would result from melting the whole snowpack instantaneously (Fierz et al., 2009).
Recent studies show negative trends in global SWE (Bormann et al., 2018; Derksen and Brown, 2012; Essery et al., 2020;
Hernández-Henríquez et al., 2015; Mortimer et al., 2020; Mudryk et al., 2017), but significant spatial variability exists: North
America shows clear negative trends in observed SWE, while negative trends are less pronounced in Eurasia (Kunkel et al.,
2016; Pulliainen et al., 2020). At mid-latitudes, SWE is more sensitive to warming than at high latitudes (Brown and Mote,
2009). Although the overall SWE trends are negative, there are also regions where SWE is observed and projected to increase:
SWE will most likely increase in northern Siberia and northern Canada (Brown and Mote, 2009; Park et al., 2012; Räisänen,
2008). Trends in seasonal snow also vary seasonally: the seasonal snow in spring is especially sensitive to warming and the
observed snow cover trends are clearly negative in both Eurasia and North America (Derksen and Brown, 2012; Essery et al.,
2020; Hernández-Henríquez et al., 2015). In winter, the observed trends are less pronounced: early winter shows even slightly
positive trends in both Eurasia and North America, while the mid-winter shows no significant trends (Hernández-Henríquez
et al., 2015).

Monitoring SWE at continental scale is only possible from satellites, yet substantial uncertainties have been reported in
satellite-based NH SWE estimates (Bormann et al., 2018; Mudryk et al., 2015). However, our knowledge of the NH SWE has
recently improved considerably, with new bias corrections which reduce the uncertainty of the SWE estimate integrated over
NH from 33% to 7.4% (Pulliainen et al., 2020). With more accurate and reliable satellite-based SWE estimates, a comparison
with the modeled SWE will also provide a more reliable analysis of the models' ability to describe the distribution of seasonal
snow.


Due to the reasons described above, it is crucial that seasonal snow is accurately described in climate models, to properly
predict the cryospheric state in future climate. However, previous studies have shown that climate models have had difficulties
in correctly reproducing the seasonal snow and its recent trends (Brutel-Vuilmet et al., 2013; Derksen and Brown, 2012;

Henderson et al., 2018; Santolaria-Otín and Zolina, 2020; Thackeray et al., 2016). Therefore, it is important to study how the
new CMIP6 (Coupled Model Intercomparison Project Phase 6) climate models can describe the seasonal snow, and where the
uncertainties and discrepancies arise.

The current paper focuses on the climatological distribution of SWE in CMIP6 models. To our knowledge, only one previous
study has compared SWE in CMIP6 models with satellite-based data: Mudryk et al. (2020) compared SWE estimates between
CMIP6 models and several observational datasets. They found that the models tend to overestimate SWE. The difference
increases in spring and is smallest in autumn and early winter. Additionally, they studied the connection between SWE and
temperature, but did not consider temperature and precipitation together. However, they stated that a coordinated analysis of
temperature and precipitation is needed to determine SWE trend drivers. Therefore, in the present study, we consider, for the
first time, the role of both temperature and precipitation for SWE differences between CMIP6 climate models and a satellite-
based dataset. Specifically, the main goals of this study are (1) to intercompare the CMIP6 and satellite-based SWE estimates
and (2) to analyze whether temperature and precipitation could explain the differences in SWE.

## 2 Data and Methods

The data of this study consist of CMIP6 climate model data (Table 1) and observational and reanalysis data (Table 2). For
CMIP6, we used monthly mean data from those models that had horizontal resolution equal to or higher than 100 km and for
which either historical or esm-hist simulations were available for download in April 2020. A total of ten models fulfilled these
criteria. The historical and esm-hist simulations extend from 1850 to 2014. In historical simulations, the $CO_2$ concentrations
are prescribed, whereas in esm-hist simulations, the models calculate the atmospheric $CO_2$ concentration interactively based
on prescribed $CO_2$ fluxes (Eyring et al., 2016). We only used relatively high-resolution models (50 or 100 km), as coarser
resolutions would differ too much from the resolution of the observational datasets, making the comparison more problematic.
In this study, we used three variables from CMIP6 models: SWE (variable "snw", unit kg m$^{-2}$), surface air temperature ("tas",
unit Kelvin), and precipitation ("pr", unit kg m$^{-2}$ s$^{-1}$). The number of ensemble members available for the chosen models varies
between 1 and 16. For simplicity, we only consider the first member of each model ensemble (r1i1p1f1) in this study. A brief
analysis showed that the differences between different ensemble members for the same model were generally smaller compared
to inter-model differences.


ESA CCI-Snow "SnowCCI" (European Space Agency Climate Change Initiative, Snow) data are based on satellite passive
microwave radiometer data and in situ data (Luojus et al., 2021; Pulliainen et al., 2020). The SnowCCI algorithm combines
microwave brightness temperature (Tb) data, observed by satellite instruments, with ground-based snow depth measurements
from the global network of synoptic weather stations (Luojus et al., 2021). The SWE estimation algorithm is based on the
difference in Tb between two frequencies (37 and 19 GHz). The ground beneath the snowpack emits microwaves, which



propagate through the snowpack, being partially absorbed during the process. The low-frequency and high-frequency signals attenuate differently as they propagate through the snowpack, which makes the difference in Tb a good indicator for estimating SWE (Cagnati et al., 2004). The attenuation is affected by snow depth, snow grain size, and snow density. The high-frequency signal attenuates more as it propagates through a deep, dense, large-grained snowpack compared to the low-frequency signal.

Thus, a large difference between high and low frequency signals indicates a high SWE (Kelly et al., 2003). The original SnowCCI algorithm combines Tb differences with in situ snow depth observations, which considerably improves SWE estimation relative to a satellite-only retrieval (Pulliainen, 2006; Takala et al., 2011).

**Table 1: CMIP6 models used in this study.**

| Institution | Model | Experiment ID | Reference |
|---|---|---|---|
| Beijing Climate Center (BCC) | BCC-CSM2-MR | historical | Wu et al. (2018a) |
| | | esm-hist | Wu et al. (2018b) |
| National Center for Atmospheric Research (NCAR) | CESM2 | historical | Danabasoglu (2019a) |
| | CESM2-WACCM | historical | Danabasoglu (2019b) |
| EC-Earth-Consortium | EC-Earth3 | historical | EC-Earth (2019a) |
| | EC-Earth3-Veg | historical | EC-Earth (2019b) |
| Geophysical Fluid Dynamics Laboratory (NOAA-GFDL) | GFDL-CM4 | historical | Guo et al. (2018) |
| | GFDL-ESM4 | historical | Krasting et al. (2018a) |
| | | esm-hist | Krasting et al. (2018b) |
| Max Planck Institute for Meteorology (MPI-M) | MPI-ESM1-2-HR | historical | Jungclaus et al. (2019) |
| Meteorological Research Institute (MRI) | MRI-ESM2-0 | historical | Yukimoto et al. (2019) |
| Seoul National University (SNU) | SAM0-UNICON | historical | Park & Shin (2019) |


A recent bias-correction method combines the original SnowCCI data with extensive ground-based snow course SWE measurements, which significantly reduces the uncertainty of NH SWE estimation (Pulliainen et al., 2020). The method decreases the uncertainty of hemisphere-mean SWE estimation from 33% to 7.4%. The bias-corrected SnowCCI data are mapped to a 25 km EASE-Grid and are available from year 1979. The data cover non-mountainous regions, and glaciers and

ice sheets are excluded. The original SnowCCI data are available around the year, while bias-corrected SnowCCI data are only available from February to May. Despite limitations in its temporal coverage, we have used the bias-corrected data in this study. We chose to do this because the bias-correction method significantly reduced the uncertainty making the data more accurate, which, in turn, makes the comparison with the models also more accurate and reliable.



Additionally, we used GPCC (Global Precipitation Climatology Centre) Version 2018 precipitation (P) data (Schneider et al., 2018) and MERRA-2 (The Modern-Era Retrospective analysis for Research and Applications, Version 2) temperature (T) data (Gelaro et al., 2017; GMAO, 2015). GPCC is a monthly precipitation product based on data from rain gauge stations, and the data are available on a 0.5-degree global grid from 1891 to the present (Schneider et al., 2018). The product agrees well with other precipitation products (Behrangi et al., 2016). The unit of CMIP6 precipitation data is kg m$^{-2}$ s$^{-1}$, whereas the unit of

GPCC data is mm month$^{-1}$, so we converted the CMIP6 data to monthly values (kg m$^{-2}$ month$^{-1}$) to make the units of the datasets equivalent to each other.

MERRA-2 is a NASA (National Aeronautics and Space Administration) atmospheric reanalysis, and it is available from year 1980. The spatial resolution of the data is 0.625°×0.5° (Gelaro et al., 2017). In this study, we have used the monthly mean 2 m

air temperature product, which agrees well with observations in the arctic and the mean values show very small biases. MERRA-2 daily temperature tends to have a cool daytime bias and warm nighttime bias (Bosilovich et al., 2015; Draper et al., 2018). However, this is not a major issue for our study because we use the monthly mean product. In addition, MERRA-2 seems to underestimate global warming trends in the last years of our study period (Gelaro et al., 2017; Simmons et al., 2017).

**Table 2: Observational and reanalysis datasets used in this study.**

| Dataset | Variable, unit | Resolution | Reference |
|---------|----------------|------------|-----------|
| SnowCCI | Snow water equivalent (SWE), mm | 25 km × 25 km, monthly | Luojus et al. (2021) |
| GPCC | Precipitation (P), mm month$^{-1}$ | 0.5° × 0.5°, monthly | Schneider et al. (2018) |
| MERRA-2 | 2 m air temperature (T), Kelvin | 0.5° × 0.625°, monthly | Gelaro et al. (2017) GMAO (2015) |

We used the nearest neighbor method to resample CMIP6, MERRA-2 and GPCC data to the 25-km equal-area projection. The SnowCCI data are only available for non-alpine regions, so we filtered out the corresponding grid cells from other datasets as

well. The difference between each model and observation was calculated by subtracting the observation value from the model value, i.e. model minus observation. We compared the differences grid cell by grid cell. Our study covered land areas north of 40° N and years between 1982 and 2014. In this study, we mainly concentrated on snow-covered areas, i.e. grid cells where SWE > 10 kg m$^{-2}$. We have also filtered out grid cells with modeled SWE above 1000 kg m$^{-2}$, as those values greatly exceed observed SWE (Stuefer et al., 2013).






We focus the analysis on two seasons: winter and spring. For the winter season, we consider only the SWE in February, since bias-corrected SnowCCI data are only available from February to May. We studied through linear regression analysis, how the difference in SWE in February between the models and the observations depends on the difference in precipitation (P) and temperature (T), averaged over the three preceding months from November to January:


$$\Delta SWE = \beta_T \Delta T_{cum} + \beta_P \Delta P_{cum} + C \qquad (1)$$

where $P_{cum}$ and $T_{cum}$ are the precipitation and temperature summed over November-January, $\beta_P$ and $\beta_T$ are the regression coefficients and C is the constant. Here, as well as in Eq. (2) below, $\Delta$ refers to the difference between the modeled value

(defined for each year separately) and the observed climatological value (averaged over the whole period considered). We used the climatological average for the observations because climate models cannot be expected to correctly simulate the weather conditions of individual years. Thus, the regression coefficients $\beta_P$ and $\beta_T$ depend only on the modeled interannual variations. The equations are presented in more detail in Appendix A.

For the spring season, we considered the monthly changes in SWE from February to March, from March to April, and from April to May. We defined the SWE change rate ($SWE_{change}$) as the difference in SWE between each month and the previous month. Positive values indicate an increase in SWE from one month to the next, and negative values a decrease. The model-minus-observation difference in $SWE_{change}$ was then regressed against the monthly difference in precipitation and temperature:

$$\Delta SWE_{change} = \beta_T \Delta T + \beta_P \Delta P + C \qquad (2)$$

For example, when considering $SWE_{change}$ from February to March, we used P and T for March. We pooled together the values of $\Delta SWE_{change}$, $\Delta P$ and $\Delta T$ for the whole spring period (February through May) to determine the regression parameters $\beta_P$ and $\beta_T$ and C. The equations are presented in more detail in Appendix A.


We included only snow-covered grid cells (SWE > 10 kg m$^{-2}$) in the analysis and calculated the linear regressions only for grid cells with at least four values available during the study period. We calculated the linear regressions for the whole study period 1982-2014, and separately for three shorter periods: 1982-1991, 1992-2001 and 2002-2014.

By substituting into Eqs. (1) and (2) the mean differences between the models and observations, it is possible to split the model biases in SWE into three components: the contribution of P ($P_C$), the contribution of T ($T_C$), and the contribution of other factors. For SWE in winter, $P_C$ and $T_C$ are:





$$P_C = \beta_P \; \Delta P_{cum,mean} \tag{3}$$

$$T_C = \beta_T \; \Delta T_{cum,mean} \tag{4}$$

Correspondingly, for $SWE_{change}$ in spring, $P_C$ and $T_C$ are:

$$P_C = \beta_P \; \Delta P_{mean} \tag{5}$$

$$T_C = \beta_T \; \Delta T_{mean} \tag{6}$$

The third component in both winter and spring is the residual term, which is the constant from the regression equation (1) or (2). This is the contribution of other factors, including for example, inaccuracies in observational datasets and model parameterizations related to, for example, snow and surface energy budget. The residual (R) gives an estimate for the SWE
bias that would remain if P and T were simulated correctly in the model.

**3 Results**

Figure 1 shows as an example the mean SWE of all CMIP6 models and SnowCCI, and ΔSWE (CMIP6-SnowCCI) in April during 1982-2014. The corresponding figures for precipitation and temperature are in the supplementary material (Fig. S1). The SWE distribution has a large spatial variability: the highest values exceed 240 kg m$^{-2}$ in both multi-model mean and
SnowCCI, and these values are found in northeastern Canada, around the Rocky Mountains, in Scandinavia, and in some parts of Siberia. Although the SWE distribution is similar for the multi-model mean and SnowCCI, the models overestimate SWE in several regions, which are mostly located in the northern parts of the study area: in northeastern Canada, northeastern Siberia, and Eurasia around 90° E. In the southern parts of the study area, the multi-model mean mainly underestimates SWE.

Figure 2 shows the monthly SWE sum of the whole study area separately for each model (grey lines), for multi-model ensemble mean (red markers) and for SnowCCI (blue markers). The blue shaded area illustrates the 7.4% uncertainty range of SnowCCI SWE estimate. There is a large variability between the models. In February, March, and April, the modeled SWE vary by a factor of two, and in May, even by a factor of three. The variability between models is notably larger than the uncertainty range of SnowCCI SWE estimate. Models reach highest SWE in March, which is consistent with observations. Overall, most models
overestimate the monthly SWE sum, and the CMIP6 multi-model ensemble mean is higher in every month except for a few years in May. While a few models underestimate the SWE sum especially in May, the majority of models overestimate the SWE sum in every month.





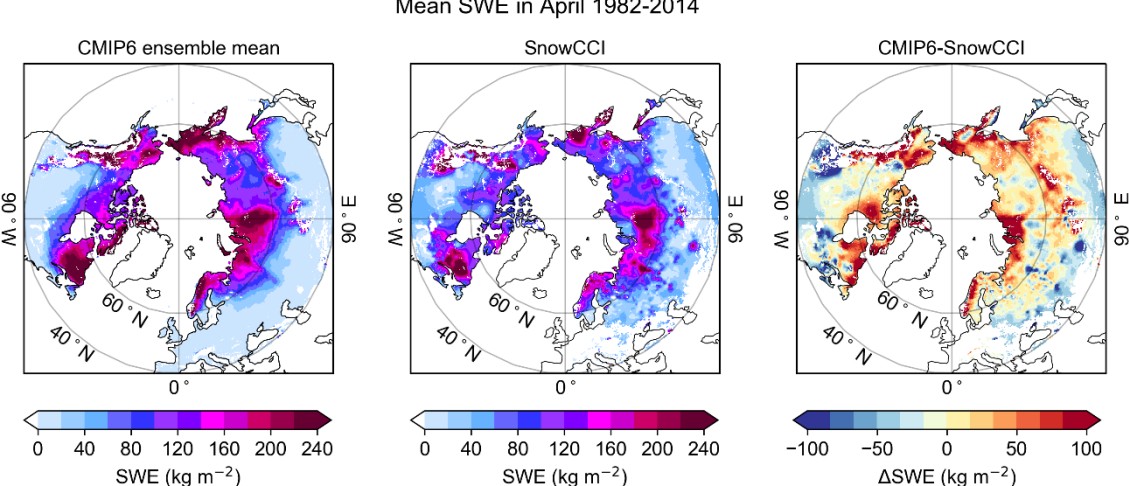

**Figure 1: Mean SWE in April for the CMIP6 multi-model ensemble mean (left), SnowCCI (middle) and the difference CMIP6-SnowCCI (right) for the period 1982-2014.**

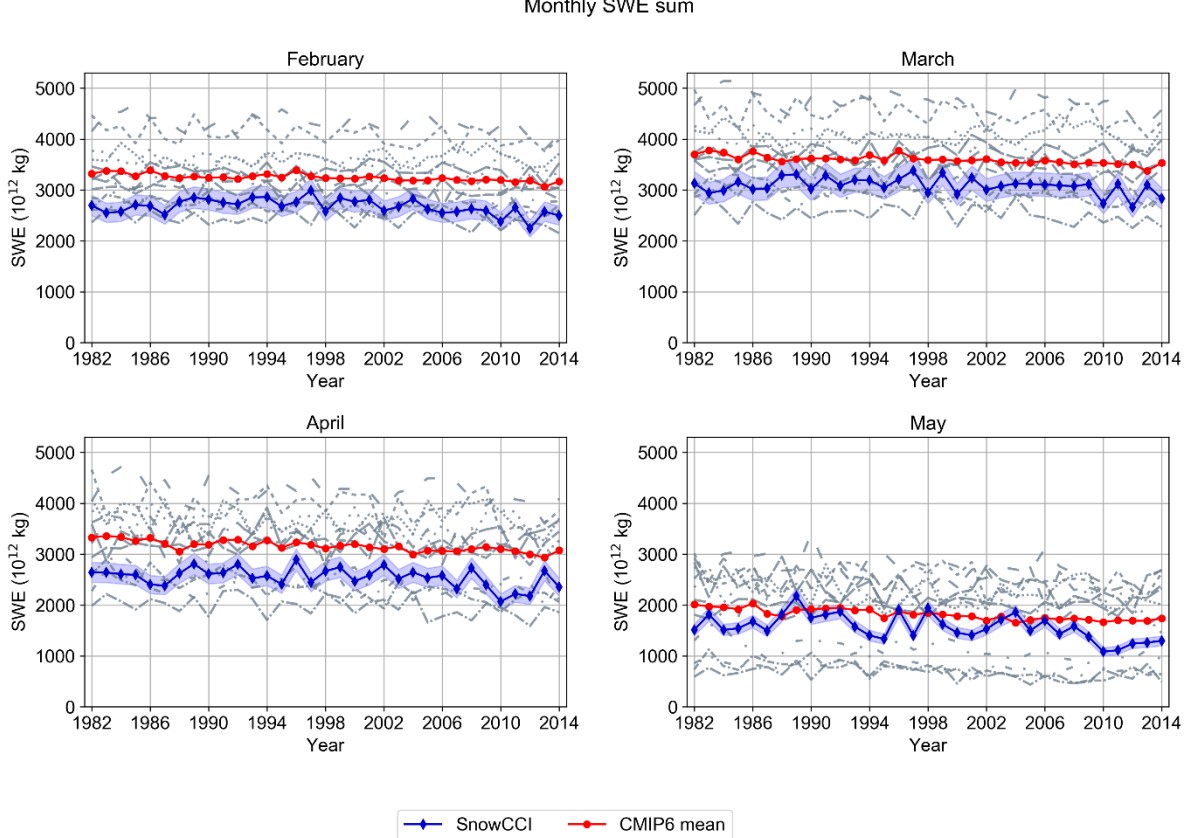

**Figure 2: Monthly SWE sum over the entire study area in February, March, April, and May separately for each CMIP6 model (grey lines), for the CMIP6 multi-model ensemble mean (red markers) and for SnowCCI (blue markers). The blue shaded area indicates the 7.4% uncertainty range of the SnowCCI data.**

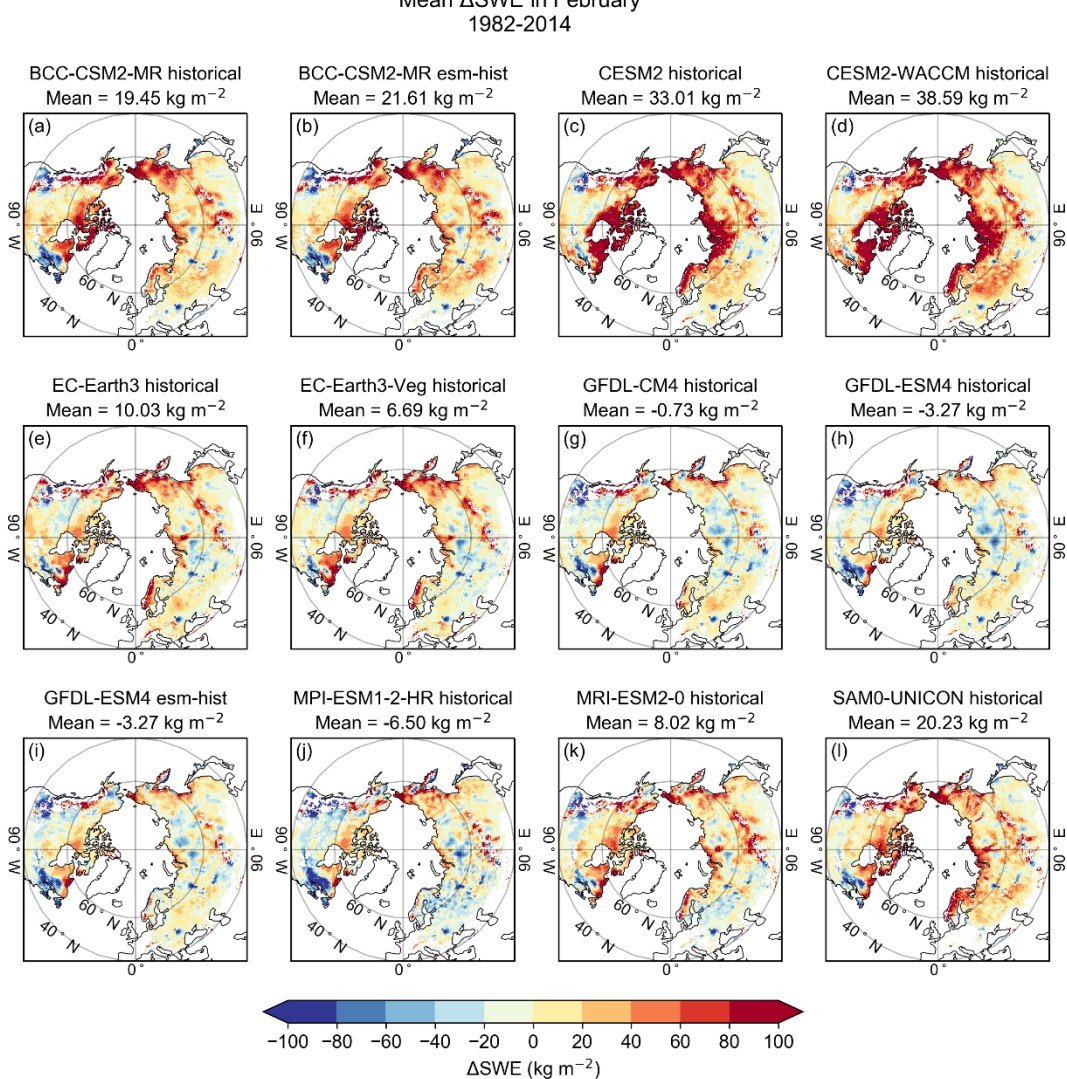

**Figure 3: Mean difference in SWE (ΔSWE, model minus observation) in snow-covered areas between each CMIP6 model and SnowCCI in February 1982-2014.**

### 3.1 SWE in winter

Figure 3 shows the mean difference in SWE in snow-covered areas between each model and SnowCCI in February for the entire study period 1982-2014. Large variability exists between the models. The areal-mean difference between the models and SnowCCI varies from about -7 kg m⁻² to over 40 kg m⁻². However, the largest negative and positive differences are well concentrated in the same areas in all models. Overall, the models tend to overestimate the SWE in the northern parts of the study area, but also in southern Siberia. The negative differences, in turn, occur mostly in the south and especially on the

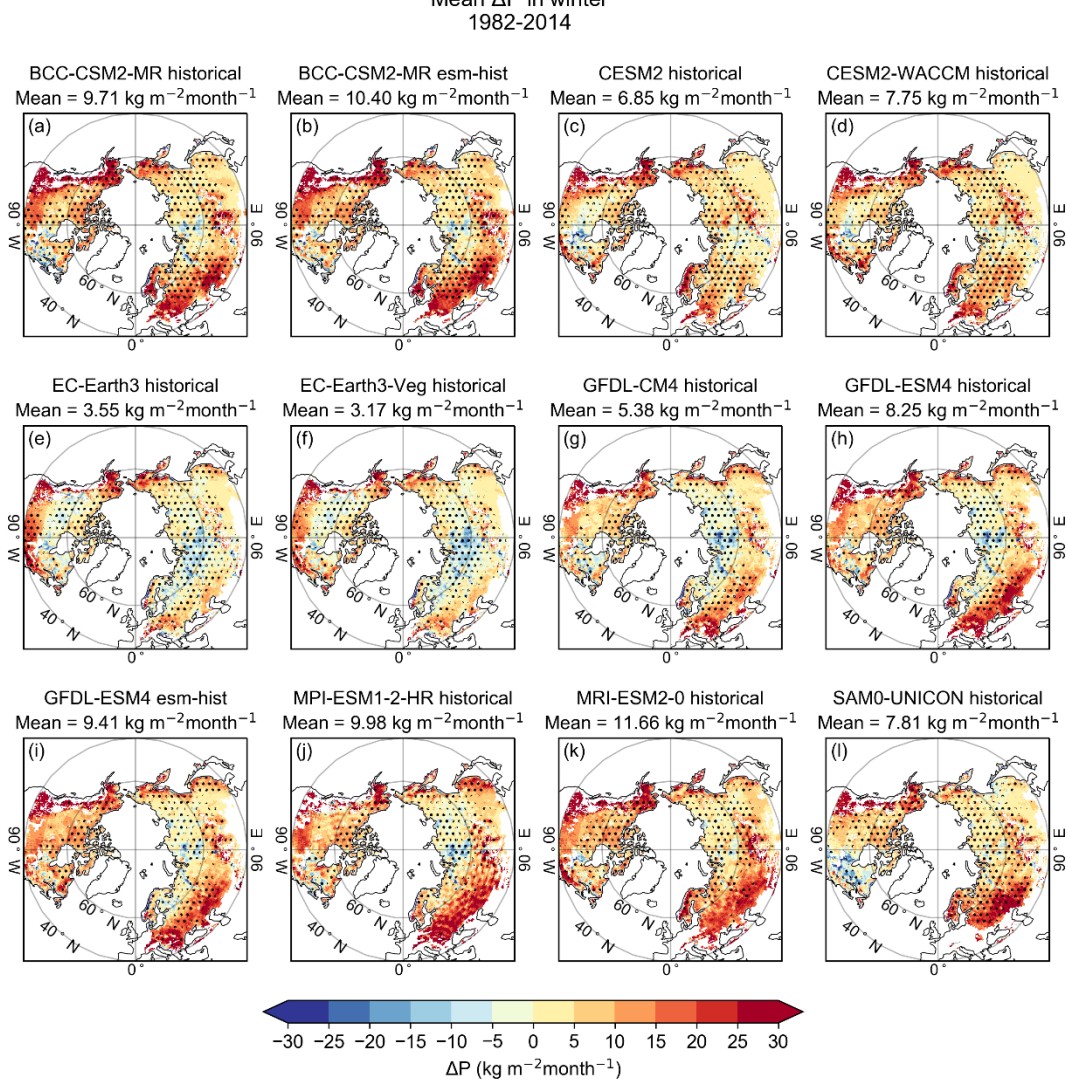

**Figure 4: Mean difference in P$_{cum}$ ($\Delta$P$_{cum}$, model minus observation) in snow-covered areas between each CMIP6 model and GPCC in winter 1982-2014. The dots indicate areas where the models either overestimate both SWE and P or underestimate both SWE and P.**

coastal areas of North America. In some models, there are negative differences also in the middle parts of Eurasia. The CESM2 and CESM2-WACCM models show the largest overestimations. For both models, the difference is very high in large regions in northern parts of North America and Eurasia. The BCC-CSM2-MR and SAM0-UNICON models also show large positive differences, which are concentrated in the same areas, although, the differences are clearly smaller than for the CESM2 and CESM2-WACCM models. In other models, the areal-mean differences are closer to 0 kg m$^{-2}$, however, regional differences

exist. Overall, the GFDL models are the most consistent with the SnowCCI data.



Figure 4 shows the mean difference in monthly $P_{cum}$ in snow-covered areas between each model and GPCC in winter for the entire study period 1982-2014. The dots indicate areas where models either overestimate both SWE and $P_{cum}$ or underestimate both SWE and $P_{cum}$, i.e. the areas where differences in $P_{cum}$ could logically explain the discrepancies in SWE. Overall, all

models overestimate precipitation in winter. The largest overestimations are mainly in southern regions and in coastal areas. There are small areas where underestimation occurs, especially in Eurasia around 90° E. In every model, there are large dotted regions where models overestimate or underestimate both SWE and $P_{cum}$. These regions are mostly in the northern parts of the study area, whereas in the south, there are more areas where SWE and $P_{cum}$ discrepancies are more often not consistent with each other.

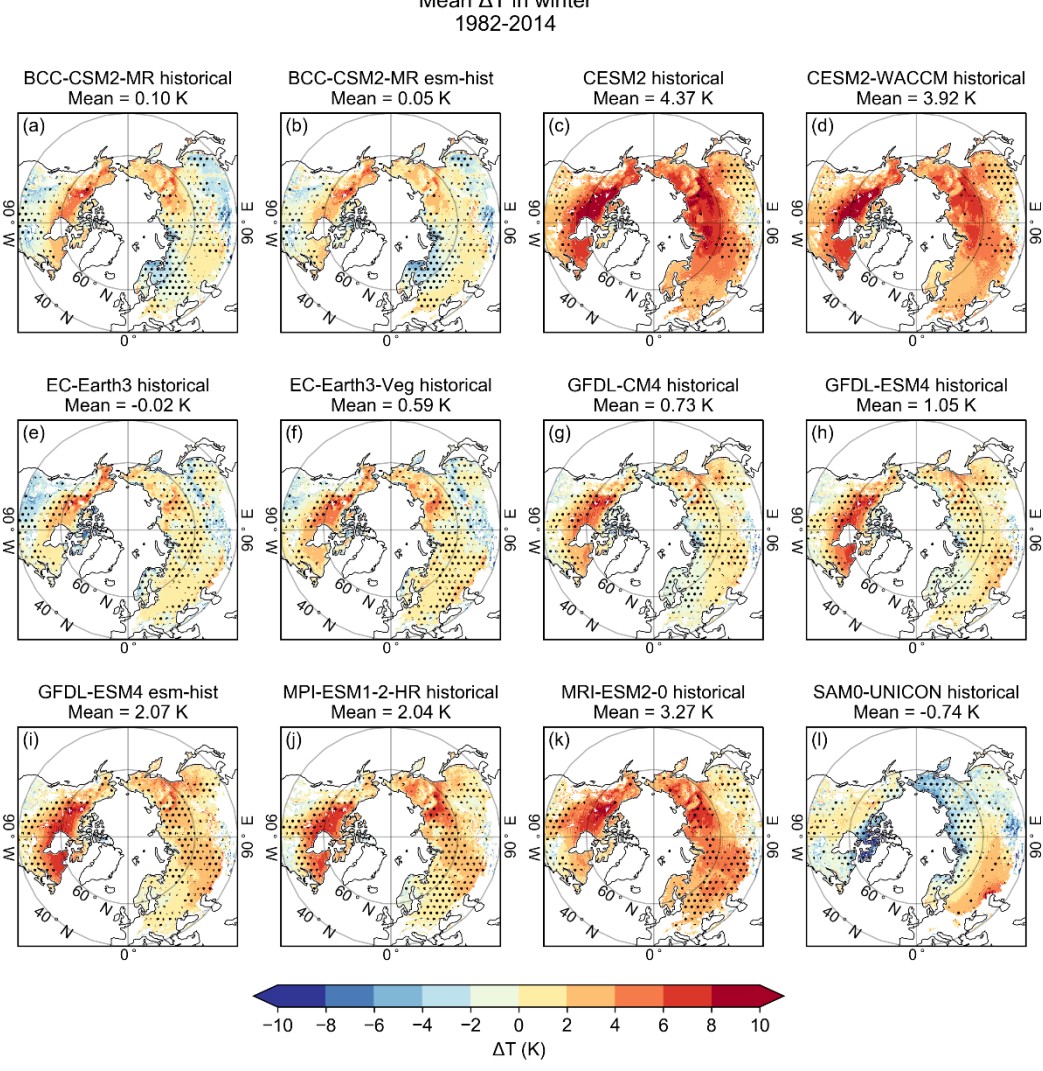

**Figure 5: Mean difference in T (ΔT, model minus observation) in snow-covered areas between each CMIP6 model and MERRA-2 in winter 1982-2014. We show ΔT (ΔT$_{cum}$ divided by 3) instead of ΔT$_{cum}$ so that the values are more intuitive and easier to interpret. The dots indicate areas where models simulate either too high SWE and too low T or too low SWE and too high T.**





Figure 5 shows the mean difference in monthly mean T in snow-covered areas between each model and MERRA-2 in winter
for the entire study period 1982-2014. We show ΔT (ΔT$_{cum}$ divided by 3) instead of ΔT$_{cum}$ so that the values are more intuitive
and easier to interpret. The differences are mostly positive; however, regional and inter-model variability exists. The CESM2
and CESM2-WACCM models generally simulate too warm temperatures and the largest positive differences are in the northern
parts of the study area. The GFDL, MPI-ESM1-2-HR and MRI-ESM2-0 models simulate too warm temperatures in the north,
while the SAM0-UNICON model, in turn, simulates too cold temperatures in the north. The BCC-CSM2-MR, EC-Earth3 and
EC-Earth3-Veg models are the most consistent with the MERRA-2 data. In every model, there are large dotted areas where
the signs of biases for T and SWE are opposite, indicating that biases in T might explain biases in SWE. However, in these
areas, the differences are mainly quite small.

The contributions to differences in SWE (ΔSWE) due to precipitation biases (P$_C$), temperature biases (T$_C$) and other factors
(residual R) are quantified using the regression equations (1), (3) and (4). To summarize their relative importance, Figure 6
shows the areal-means of the absolute values of ΔSWE, P$_C$, T$_C$, and R. The CESM2 and CESM2-WACCM models show
largest ΔSWE, whereas in other models, ΔSWE is clearly smaller. In all models, the contribution of P on ΔSWE is clearly
larger than the contribution of T. However, the residual is also typically large, indicating that P and T cannot explain the SWE
biases alone. This implies that observational uncertainty or model structural factors play a considerable part in the observed
SWE differences. The variability in these parameters between the decadal subperiods and the full three-decade analysis period
was slight, indicating consistent behavior across time in both models and observations.

Figure 7 shows the spatial distribution of the contributions of P and T and the residual for each model for the entire study
period 1982-2014. The regression parameters R$^2$, β$_P$ and β$_T$ are shown in the supplementary material (Figure S2). Also, the
contributions of P and T and the residual calculated for the shorter time periods are in the supplementary material (Figures S3-
S5). Fig. 7 shows that overall, the contribution of P is larger than the contribution of T, as Fig. 6 already indicated. P contributes
to ΔSWE especially over northern and coastal regions, with fairly similar patterns for all models considered. The regression
coefficient β$_P$ also shows large values (β$_P$ ≈ 1) especially in the northern and middle parts of both continents (Fig. S2), with
relatively small inter-model variations. This is consistent with the expectation that in cold regions, an increase of precipitation
translates into a similar increase in SWE.

The contribution of T is mostly very weak (Fig. 7); however, for some models, T shows stronger contribution especially over
western parts of Eurasia and over northeastern Canada. The regression coefficient β$_T$ is mostly negative or very close to zero
(Fig. S2). The negative correlation is strongest in Europe and southern parts of North America and Eurasia. In these regions,
the temperature is close to 0 °C in winter, which makes temperature an important driver of the SWE. Northern Canada and
Siberia, in turn, show areas with positive correlation, meaning that warmer temperatures cause higher SWE. Studies have





shown that when the temperature rises due to climate change, the winter precipitation will also increase (Brown and Mote, 2009; Park et al., 2012; Räisänen, 2008). In the south, warming temperatures will shift winter precipitation from snow to rain. In the north, in turn, temperature will stay below 0 °C despite the warming, which will lead to an increase in snowfall in coldest

regions of NH, and therefore, to an increase in SWE. This phenomenon is most likely seen here as well; warmer temperatures in the models will increase winter precipitation, resulting in too high SWE in the models. It should be noted that since Eq. (1) treats $\Delta T_{cum}$ and $\Delta P_{cum}$ as independent variables, a positive correlation between the variables means that their contributions to $\Delta$SWE cannot be fully separated.

The residual shows large spatial and inter-model variability (Fig. 7). Especially for the CESM2 and CESM2-WACCM models, the residual shows very large positive values. These large positive residuals are mainly concentrated in the same areas where the models clearly overestimate SWE (Fig. 3). This indicates that, for these models, the large SWE differences in these areas are mainly caused by some other factors than P or T. For other models, the residual shows both positive and negative values across the study area.


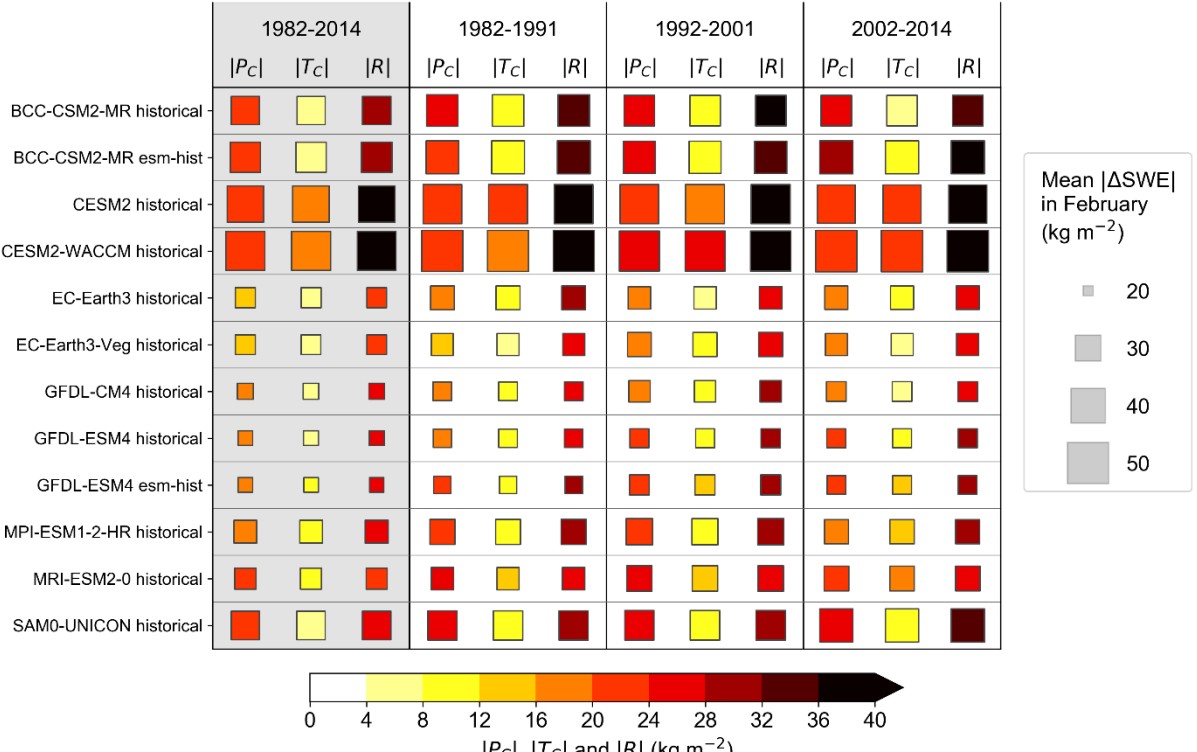

**Figure 6: The areal-means of the absolute values of $\Delta$SWE, $P_C$, $T_C$, and residual R calculated for the entire study period 1982-2014 (left column, shaded with grey) and for three shorter time periods (1982-1991, 1992-2001, and 2002-2014) for each model in winter. The size of the square indicates the absolute value of $\Delta$SWE of that time period and model, and the color of the square indicates the**

**absolute value of $P_C$, $T_C$, and R.**



**Figure 7: Spatial distribution of the P contribution, the T contribution, and the residual for each model in winter 1982-2014.**

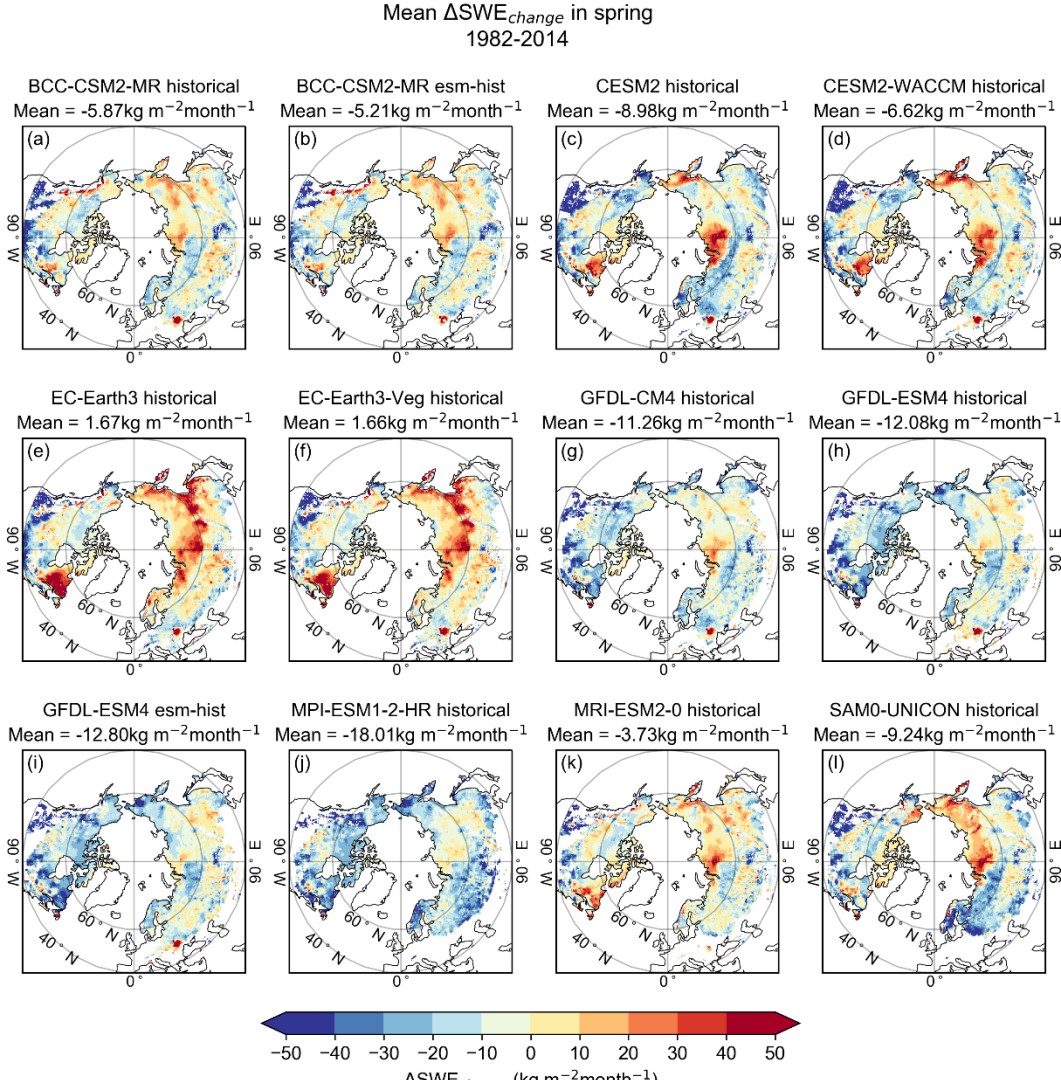

**Figure 8: Mean difference in the monthly SWE change (ΔSWE$_{change}$, model minus observation) in spring between each CMIP6 model and SnowCCI for the period 1982-2014.**

## 3.2 Monthly SWE change in spring

Figure 8 shows the mean difference in monthly SWE change (ΔSWE$_{change}$) in spring between each model and SnowCCI for the whole study period 1982-2014. Positive ΔSWE$_{change}$ means that snow melts more slowly in the models compared to SnowCCI, and negative ΔSWE$_{change}$ indicates that snow melts faster in models, respectively. The areal-mean ΔSWE$_{change}$ is mainly negative in every model, which means that snow melts generally faster in the models compared to SnowCCI. However, there is a large spatial variability in every model and inter-model variability is also large. In the CESM2 and CESM2-WACCM



models, three areas show distinctly positive $\Delta SWE_{change}$: northeastern Canada, northern Siberia, and eastern Siberia. In all these areas, the SWE difference in February (Fig. 3) was already clearly positive, meaning that these models greatly overestimate SWE in these areas also in spring. The EC-Earth3 and EC-Earth3-Veg models show clear positive $\Delta SWE_{change}$ in northeastern

Canada but also in Eurasia. The area with positive differences in Eurasia is very extensive and differs notably from the other models. The SAM0-UNICON model also shows positive values in northern Siberia. The GFDL and MPI-ESM1-2-HR models, in turn, show large areas with negative $\Delta SWE_{change}$ in northern Canada and in eastern parts of Siberia, which differs from the other models. Overall, the model-minus-observation differences in SWE change rate in spring (Fig. 8) show larger inter-model variations than the corresponding differences in SWE in February (Fig. 3).

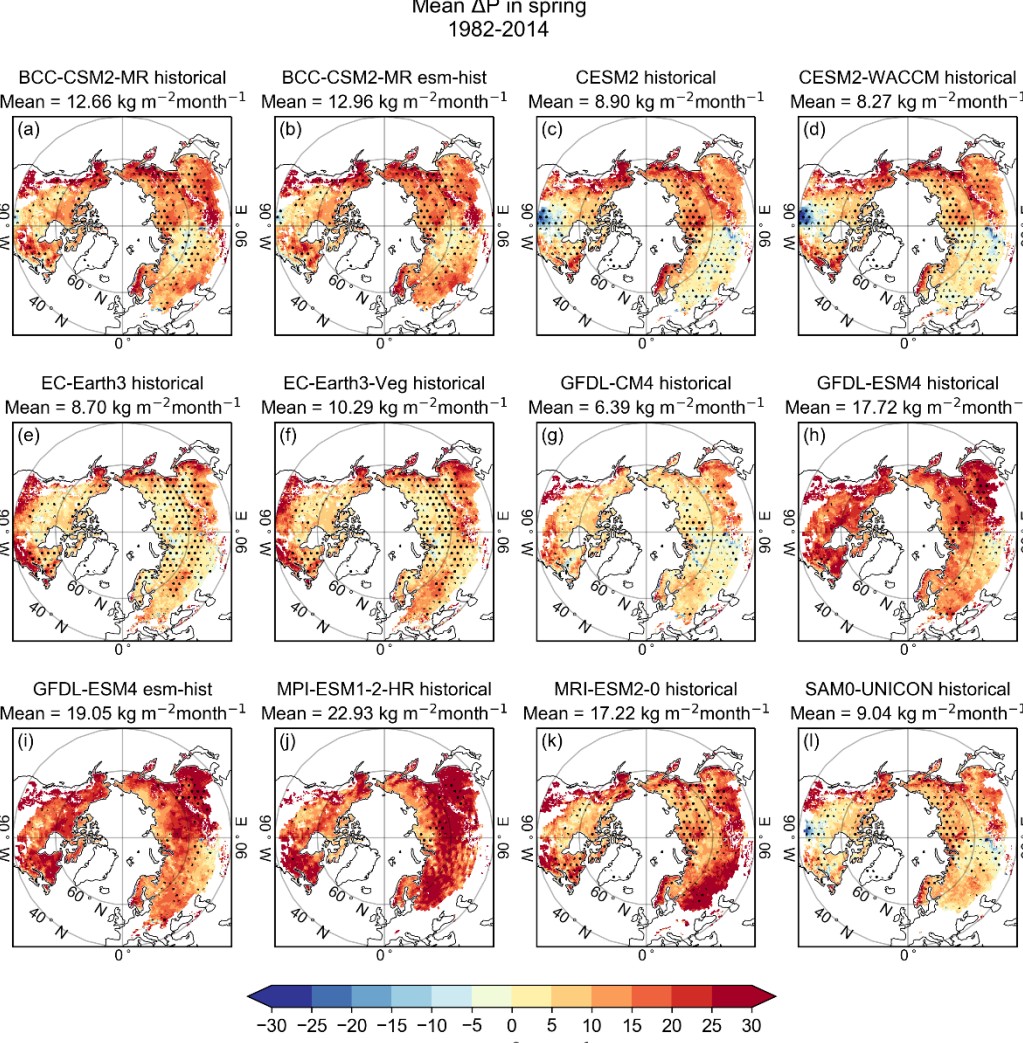


**Figure 9: Mean difference in P (ΔP, model minus observation) in snow-covered areas between each model and GPCC in spring 1982-2014. The dots indicate areas where models either overestimate both SWE$_{change}$ and P or underestimate both SWE$_{change}$ and P.**





Figure 9 shows the mean difference in P over snow-covered areas between each model and GPCC in spring for the entire study

period 1982-2014. As in winter (Fig. 4), models on average overestimate precipitation in spring as well. The largest overestimations occur mainly in southern regions and in coastal areas. The regions with mutually biases in P and $SWE_{change}$ show large inter-model variability, and they are less extensive than in winter (Fig. 4). This indicates that precipitation is not as important factor in spring than it is in winter.

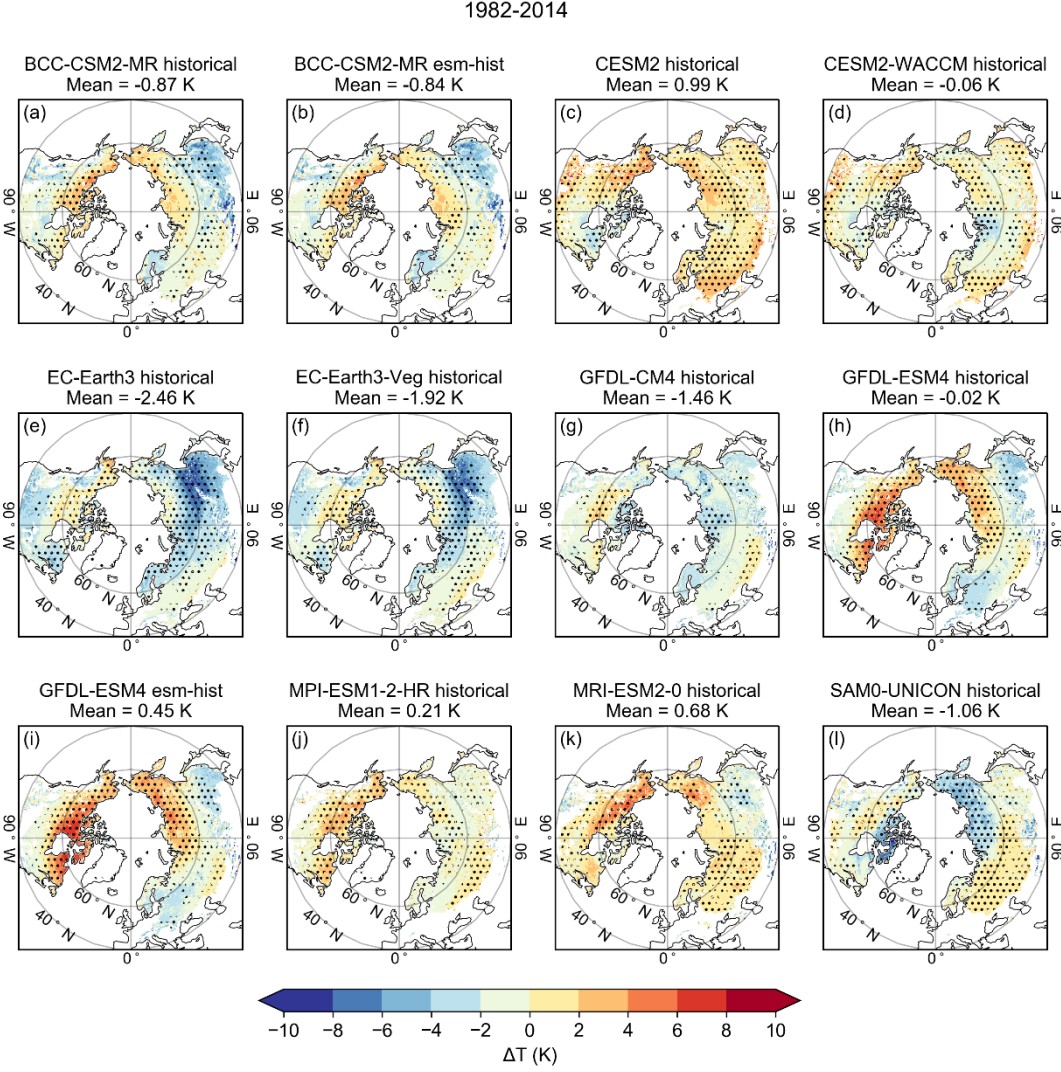

**Figure 10: Mean difference in T ($\Delta$T, model minus observation) in snow-covered areas between each CMIP6 model and MERRA-2 in spring 1982-2014. The dots indicate areas where models simulate either too positive $SWE_{change}$ and too low T or too negative $SWE_{change}$ and too high T.**





Figure 10 shows the mean difference in T in snow-covered areas between each model and MERRA-2 in spring for the entire
study period 1982-2014. There is a large spatial variability within each model but also the variability between models is very
clear. The BCC-CSM2-MR, GFDL-ESM4, MPI-ESM1-2-HR and MRI-ESM2-0 models show a warm bias in the northern
parts of the study area, whereas SAM0-UNICON shows a cold bias in the same area. The EC-Earth3 and EC-Earth3-Veg
models, in turn, have a cold bias in eastern Eurasia near 60° N, which clearly differs from the other models. The sizes and
locations of the dotted areas (i.e., areas with mutually consistent biases in T and $SWE_{change}$) vary greatly between models.
Especially in the GFDL-CM4 model, the size of these areas is small, while in most of the models, the dots cover majority of
the study area. Furthermore, in those regions where the biases in T and $SWE_{change}$ are consistent in spring, the cold or warm
temperature biases are typically relatively large, when compared with the corresponding biases in winter  (Fig. 5). This
indicates that biases in T are more important driver of biases in SWE in spring than in winter.

Figure 11 summarizes the areal-means of the absolute values of $\Delta SWE_{change}$, the contribution of P ($P_C$), the contribution of T
($T_C$), and the contribution of other factors (the residual R). In all models, the residual is larger than $P_C$ or $T_C$. This suggests that
overall, the biases in snow melt rate in spring are dominated by other factors than T or P. The contributions of P and T are
quite similar in magnitude but varies between models. The EC-Earth3 model stands out from the other models, as $T_C$ is larger
than in the other models. None of the variables shows a large dependence on the period considered.


Figure 12 shows the spatial distribution of the contributions of P and T and the residual for each model for the entire study
period 1982-2014. The regression parameters $R^2$, $\beta_P$ and $\beta_T$ are displayed in the supplementary material (Figure S6). Also, the
contributions of P and T and residual calculated for the shorter time periods are in the supplementary material (Figures S7-
S9). P contributes to $\Delta SWE_{change}$ mostly in Alaska and northern Siberia, but the effect varies between models. Furthermore,
even though T showed clear warm and cold biases in many areas (Fig. 10), the contribution of T is mostly quite weak, because
of the small regression coefficient $\beta_T$ (Fig. S6). However, exceptions exist; especially, the EC-Earth3 and EC-Earth3-Veg
models stand out, as in Eurasia, there is a large area where a negative bias in T (Fig. 10) contributes substantially to a positive
bias in $SWE_{change}$ (Fig. 8). Also, for the CESM2 and GFDL-ESM4 models, T shows a stronger contribution over northern parts
of North America. Overall, however, the contributions of both P and T are small compared to the residual R, which is consistent
with Fig. 11. This indicates that other factors than T or P are the dominant drivers for the $SWE_{change}$ discrepancies. The residual
is mostly negative in all the models, which means that snow would melt too fast in the models, if T and P were simulated
correctly. This issue is discussed further in Sect. 4.



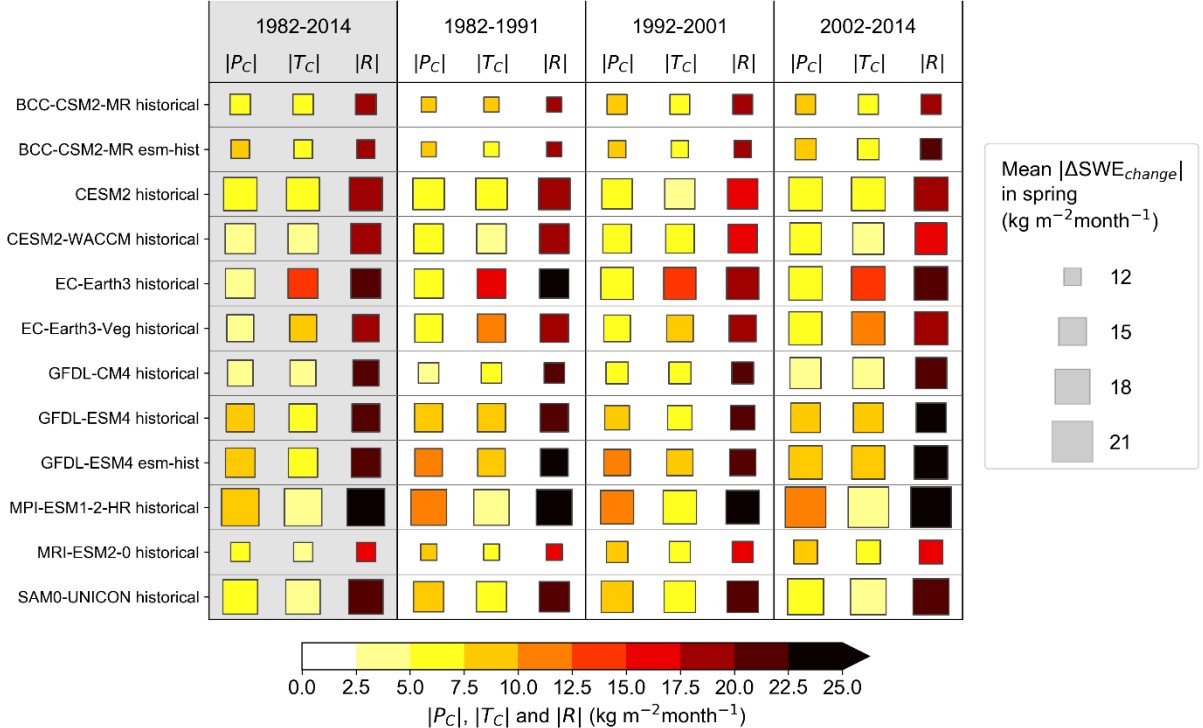

**Figure 11: The areal-means of absolute values of ΔSWE_change, P_C, T_C, and residual R calculated for the entire study period 1982-2014 (left column, shaded with grey) and for three shorter time periods (1982-1991, 1992-2001, and 2002-2014) for each model in spring. The size of the square indicates the mean absolute value of ΔSWE_change of that time period and model, and the color of the square indicates the mean absolute value of P_C, T_C, and R.**

## 4 Discussion

We have evaluated NH SWE in CMIP6 models with satellite-based SnowCCI data for the period 1982-2014. While SWE in CMIP6 models has been previously studied by Mudryk et al. (2020), this is to our knowledge the first study to analyze CMIP6 SWE together with temperature and precipitation. In addition, the recent bias-correction method has significantly narrowed down the confidence limits of the NH SWE estimate, which makes the comparison more accurate and provides a more reliable analysis of the models' ability to describe the snow cover. The continuous development of climate models is crucial, so that the changing climate can be simulated as reliably as possible.

The analysis shows that CMIP6 models overestimate SWE with a few exceptions (Fig. 2), which is consistent with the previous study (Mudryk et al., 2020). The overestimation of SWE was also observed in CMIP5 models (Santolaria-Otín and Zolina, 2020). The NH SWE sum reaches its peak value in March, but the peak values are overestimated by most of the models. As the spring advances, the variability between models increases, and some of the models clearly overestimate the SWE in May.



**Figure 12: Spatial distribution of the P contribution, the T contribution, and the residual for each model in spring 1982-2014.**



In some models, in turn, snow melts faster than in the observations and they underestimate SWE in late spring. This is also shown in Fig. 3 and 8: in winter, the differences in SWE are mainly positive (Fig. 3), while in spring, there are large differences in snow melt rates between the models (Fig. 8).


In this study, we have analyzed whether differences in T and P between models and observations could explain the corresponding differences in SWE. The analysis focused, on one hand, on the SWE in February and, on the other hand, on the SWE change rate ($SWE_{change}$) during the spring from February to May. Using linear regression, model biases in SWE ($\Delta SWE$) and $SWE_{change}$ ($\Delta SWE_{change}$) were divided into three components: the contribution of P, the contribution of T and the

contribution of other factors.

In winter, the models mostly overestimate SWE (Fig. 3), but spatial and inter-model variability exists. The overestimations are mostly concentrated in the same areas in all models, but the magnitude varies greatly between the models. The models also overestimate precipitation in winter (Fig. 4), suggesting that overestimated SWE is caused by the overestimated P. The

regression coefficient $\beta_P$ also shows very clear correlation between $\Delta SWE$ and $\Delta P_{cum}$ (Fig. S2). Therefore, P clearly contributes to $\Delta SWE$, whereas the contribution of T is substantially smaller (Fig. 7). This result is consistent with the expectation that precipitation is needed to initiate the snow cover. In other words, even too cold temperatures cannot cause too high SWE without precipitation. The link between $\Delta T_{cum}$ and $\Delta SWE$ is strongest in the warmest regions of the study area (Fig. 7 and S2). Especially in the coldest regions, where T is well below 0 °C, variations in T do not significantly affect the amount of snow

on the ground. In regions where T is closer to 0 °C in winter, T plays a more significant role and has a greater impact on SWE. This physically reasonable behavior suggests that climate models might be able to simulate SWE trends in the warming climate correctly, even if SWE itself is not reproduced accurately.

In spring, $\Delta SWE_{change}$ and $\Delta T$ show quite similar patterns in many models (Fig. 8 and 10), which indicates that biases in T

affect the biases in $SWE_{change}$. This result is to be expected because the increase in T is the main factor that causes snow to melt in spring. A relationship between temperature and snow cover in spring has also been observed in CMIP5 models (Brutel-Vuilmet et al., 2013; Mudryk et al., 2017). The CMIP5 models have been found to underestimate the observed trend towards a reduced snow cover extent due to an underestimation of the spring warming trend (Brutel-Vuilmet et al., 2013).

Even though $\Delta SWE_{change}$ and $\Delta T$ are quite consistent with each other, the contribution of T is not very strong (Fig. 12), because the regression coefficient $\beta_T$ is small (Fig. S6). Several factors can weaken the regression coefficient $\beta_T$. The analysis shows that in spring, the relationship between $\Delta T$ and $\Delta SWE_{change}$ is strongest in western parts of Eurasia and in southern parts of North America (Fig. S6). These are regions with the earliest snow melt onset. In these areas, T is the dominant factor causing snow to melt throughout the spring season. In the northernmost parts of the study area, the melt season begins later, so that

early spring still belongs to the snow accumulation season. As a result, P may still be the dominant factor influencing





$\Delta SWE_{change}$ in early spring, and T will become a more significant factor later in the spring. This can reduce the correlation and affect the linear regression parameters in the northern parts of the study area. Additionally, if SWE in winter is simulated incorrectly, this can affect the melt rates in spring, as there can be too little or too much snow that can melt.

In both winter and spring, the residual term R of the regressions is also significant (Figs. 6, 7, 11 and 12). This means that the model biases in SWE in winter and $SWE_{change}$ in spring cannot be entirely explained by the biases in P and T, but other factors also contribute to these biases. These factors may include inaccuracy in the model parameterizations related to snow and surface energy budget, but also inaccuracy in the observational datasets.

The residual term R is particularly pronounced in spring, when it is typically larger than either the contribution of P or T. Interestingly, the residual is mostly negative (Fig. 12). The negative residual means that if P and T were simulated correctly in the models, snow would melt too fast in spring. While understanding the origins of this bias would be worth a separate study, a previous study with ECHAM5 (Räisänen et al. 2014) is of interest here. ECHAM5 is a predecessor of the atmospheric part of MPI-ESM1-2-HR, for which the residual R in Fig. 12 is especially strongly negative. This is consistent with the finding that
in ECHAM5 snow melted generally too fast in spring, despite a cold bias in T (Räisänen et al. 2014). A major factor for this was that T was not calculated separately for snow-covered and snow-free parts of the grid cell. Because of that, T was not able to rise above 0 ℃ if there was snow left in the grid cell, and, therefore, a too large fraction of the available energy was used in melting the snow (Räisänen et al., 2014). The parameterization of the surface albedo is another factor that may influence the snowmelt rate in spring. A too low (or high) surface albedo would speed up (or delay) the snowmelt. As a detail, the only
region in which MPI-ESM1-2-HR displays a positive residual in Fig. 12 is southeastern Siberia. In this very region, ECHAM5 featured delayed snowmelt, related to overly high albedo in the presence of vegetation over snow (Räisänen et al. 2014). While the specific mechanisms leading to too fast snow melt might differ in different models, the example of ECHAM5 highlights the importance of the treatment of surface energy budget in the presence of snow.

All the observational datasets have uncertainties that can also affect the results. With the bias-correction method, SWE data is more accurate than before, but the uncertainty in hemisphere-mean values is still 7.4%. There are also uncertainties associated with the GPCC and MERRA-2 datasets that can cause errors in the differences between models and observations; for example, MERRA-2 underestimates global warming trends in the last years of our study period compared to other reanalyses (Gelaro et al., 2017; Simmons et al., 2017). Snow cover in spring is especially sensitive to warming (Hernández-Henríquez et al., 2015)
and, therefore, the uncertainties in MERRA-2 can affect the results especially in spring.



## 5 Conclusions

We have intercompared NH SWE estimates between CMIP6 models and satellite-based SnowCCI data, and studied whether differences in precipitation (P) or temperature (T) between models and observations could explain the differences in SWE. Our study covered land areas north of 40°N and years between 1982 and 2014. We analyzed separately the SWE in winter (in
February) and the SWE change rate in spring ($SWE_{change}$ from February to May). Using regression analysis, we divided the difference in SWE between model and observation ($\Delta SWE$ and $\Delta SWE_{change}$, model minus observation) into three components: the contribution of P, the contribution of T and the contribution of other factors, such as deficiencies in model parameterizations or inaccuracies in the observational datasets. The main findings in our study are as follows:

- The models generally overestimate SWE, but large variability exists between models. The largest overestimations occur mainly in the northernmost parts of both Eurasia and North America. In winter, the overestimated SWE is mainly concentrated in the same areas in every model, but the magnitude differs between the models. In spring, the snow melt rates vary clearly between the models.

- In winter, the differences in SWE can be explained mostly with differences in P. The contribution of T is clearly
smaller than that of P. This is in line with the expected results, as even too cold temperatures cannot cause too high SWE without precipitation. However, other factors contribute to SWE discrepancies as well.

- In spring, T and P explain partly the differences between modeled and observed $SWE_{change}$. Especially cold or warm biases often co-occur with large $SWE_{change}$ differences, but large spatial and inter-model variability exists. The importance of T in explaining $SWE_{change}$ discrepancies during spring is to be expected, because the increase in T is
the main factor that causes snow to melt as spring progresses. Yet it should be noted that the contribution of other factors, such as observation uncertainty or deficiencies in model parameterizations, is more significant in spring than in winter.

Overall, the study showed that the models still need to be improved to accurately describe SWE. However, the analysis also
showed that there is a link between T and SWE, especially in the warmer regions of the study area, suggesting that climate models may be able to simulate SWE trends in a warming climate correctly, even if SWE itself is not accurately reproduced. Uncertainties in the observational data also cause uncertainties in the analysis, so by improving the observational data, we can study the models' ability to describe the snow cover more reliably and, thus, further improve the models.

## Appendix A: The equations for calculating the differences between models and observations and the linear regressions

The steps for calculating the differences in SWE, T and P between models and observations, and subsequently the linear regressions in winter are as follows:





1. We calculated cumulative T ($T_{cum}$) and P ($P_{cum}$) from November to January for each model and for the observational datasets:

$$T_{cum,\ model} = T_{Nov} + T_{Dec} + T_{Jan} \tag{A1}$$


$$T_{cum,\ obs} = T_{Nov} + T_{Dec} + T_{Jan} \tag{A2}$$

$$P_{cum,\ model} = P_{Nov} + P_{Dec} + P_{Jan} \tag{A3}$$

$$P_{cum,\ obs} = P_{Nov} + P_{Dec} + P_{Jan} \tag{A4}$$

2. We calculated the difference in cumulative T ($\Delta T_{cum}$) and P ($\Delta P_{cum}$) between each model and observations:


$$\Delta T_{cum} = T_{cum,\ model} - T_{cum,\ obs} \tag{A5}$$

$$\Delta P_{cum,} = P_{cum,\ model} - P_{cum,\ obs} \tag{A6}$$

3. We calculated the difference in SWE ($\Delta SWE$) in February between each model and SnowCCI:

$$\Delta SWE = SWE_{model} - SWE_{obs} \tag{A7}$$


4. We calculated the linear regression for the differences using the ordinary least squares method:

$$\Delta SWE = \beta_T \Delta T_{cum} + \beta_P \Delta P_{cum} + C \tag{A8}$$

where $\beta_T$ and $\beta_P$ are the regression coefficients, and C is the constant.

The steps for calculating the differences in $SWE_{change}$, T and P between models and observations, and subsequently the linear regressions in spring are as follows:

1. We calculated monthly change in SWE ($SWE_{change}$) for each model and for SnowCCI:

$$SWE_{change,1} = SWE_{Mar} - SWE_{Feb} \tag{A9}$$


$$SWE_{change,2} = SWE_{Apr} - SWE_{Mar} \tag{A10}$$

$$SWE_{change,3} = SWE_{May} - SWE_{Apr} \tag{A11}$$

2. We calculated the differences in monthly $SWE_{change}$ ($\Delta SWE_{change}$) between each model and SnowCCI:

$$\Delta SWE_{change} = SWE_{change,1,model} - SWE_{change,1,obs} \tag{A12}$$


$$\Delta SWE_{change} = SWE_{change,1,model} - SWE_{change,1,obs} \tag{A13}$$

$$\Delta SWE_{change} = SWE_{change,1,model} - SWE_{change,1,obs} \tag{A14}$$

3. We calculated the differences in T ($\Delta T$) and P ($\Delta P$) between each model and the observations in March, April, and May:

$$\Delta T = T_{Mar,model} - T_{Mar,obs} \tag{A15}$$



500   $\Delta T = T_{Apr,model} - T_{Apr,obs}$    (A16)

$\Delta T = T_{May,model} - T_{May,obs}$    (A17)

$\Delta P = P_{Mar,model} - P_{Mar,obs}$    (A18)

$\Delta P = P_{Apr,model} - P_{Apr,obs}$    (A19)

$\Delta P = P_{May,model} - P_{May,obs}$    (A20)


3. We pooled together the values of $\Delta SWE_{change}$, $\Delta P$ and $\Delta T$ for the whole spring period (February through May) and calculated the linear regression for the differences using the ordinary least squares method:

$\Delta SWE_{change} = \beta_T \Delta T + \beta_P \Delta P + C$    (A21)

where $\beta_T$ and $\beta_P$ are the regression coefficients, and C is the constant.

**Data availability**

The CMIP6 model data are available at the Earth System Grid Federation (https://esgf-node.llnl.gov/search/cmip6/). The SnowCCI data are available at Globsnow data archive (https://www.globsnow.info/). The MERRA-2 data are available at the Modeling and Assimilation Data and Information Services Center (https://disc.gsfc.nasa.gov/). The GPCC data are available at DWD website (https://opendata.dwd.de/climate_environment/GPCC/html/fulldata-monthly_v2018_doi_download.html).

**Author contribution**

KK performed the analysis and produced the figures with substantial contributions from AR and PR. KL provided the SnowCCI data. KK wrote the original draft. All authors contributed to manuscript review and editing.

**Acknowledgements**

The work of all authors has been funded by the Academy of Finland, Decision 309125.

**Competing interests**

The authors declare that they have no conflict of interest.





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
