# Peer review of "Evaluation of Northern Hemisphere snow water equivalent in CMIP6 models during 1982-2014"

_The Cryosphere, 2021_

## Author Comment (AC1)

The review comments are shown in black, the author responses in blue and text from revised manuscript in *blue italics*.

We would like to thank the reviewer for reviewing our manuscript. We appreciate all the comments and have revised our manuscript according to them. Please find below our responses to the comments.

**Referee #1**

**Major comments**

Model selection: The decision to limit analysis to a subset of high resolution GCMs seems somewhat arbitrary and limits the paper's value. This decision should be better justified in the text. For example,
10    the authors could show a comparison of winter SWE in high vs low resolution models as supplemental material. Otherwise, the authors should consider adding a few of the HighResMIP historical simulations (https://gmd.copernicus.org/articles/9/4185/2016/gmd-9-4185-2016.pdf) to their analysis so as to increase the ensemble size.

15    We have now downloaded all the available CMIP6 models and compared SWE in the high resolution (100 km) and low resolution (250 or 500 km) models. We compared the monthly SWE sums over the entire study area (Fig. 1), following Fig. 2 in the submitted manuscript. Fig. 1 shows that the low-resolution models (thin lines) do not significantly differ from the high-resolution models (dashed lines); the SWE sum is mostly in the same range in both resolution groups. Also, the mean values for both
20    high-resolution and low-resolution models are very close to each other. Therefore, we have performed the subsequent detailed analysis only for the high-resolution models to keep the number of models reasonable and to only consider models with spatial resolutions which are reasonably comparable with e.g., the SnowCCI data. The number of high-resolution models in the analysis has also slightly increased from the previous version, as we now included all models that were available for downloading in August
25    2021. Fig. 1 shows that there are two low-resolution models that show very high SWE sum values in every month, which are clear outliers. These outliers are "GISS-E2-1-G historical" and "GISS-E2-1-G-CC esm-hist" and we have found that the anomalous values are due to very high SWE in the mountainous areas.

30    Following Referee #2's comments, we have also added mountainous regions to the analysis and, therefore, the SWE sum values are somewhat larger than in the previously submitted version. As the SnowCCI data are available only for non-mountainous areas, we have used MERRA-2, Brown and Crocus v7 datasets to fill the missing SWE over mountainous areas. We calculated the mean SWE of these three datasets for each grid cell that were defined as mountainous in SnowCCI.

35

[Figure]

**Figure 1. Monthly SWE sum over the entire study area in February, March, April, and May separately for each high-resolution (100 km) CMIP6 model (grey dashed lines), for each low-resolution (> 100 km) CMIP6 model (thin solid lines), for the high-resolution CMIP6 multi-model ensemble mean (red markers), for the low-resolution CMIP6 multi-model ensemble mean (purple markers) and for the SWE reference data (blue markers). The blue shaded area indicates the 7.4% uncertainty range of the SWE reference data.**

Interpretation of results: The authors point out discrepancies between models and observations but offer little commentary on what could be driving biases in specific GCMs. For example, they discuss a cold bias in the EC-Earth models as unique to the ensemble but fail to connect this to the fact that EC-Earth is the largest outlier in terms of snow cover extent among CMIP6 models (Mudryk et al. 2020). More insight could also be added when discussing the CESM models, which feature anomalous winter SWE.

We have rewritten the Discussion and discuss more about both EC-Earth3 and the CESM2 models. We added discussion about the snow cover extent in EC-Earth3 models. We also contacted the CESM2 model developers to get more information about the anomalous SWE in the CESM2 models. A unique feature within CESM2, which should be considered, is that the model allows for a very large maximum SWE (10 m). This is done to enable the simulation of firn production over ice sheet regions, but it is possible that SWE can get very high in other cold regions as well (van Kampenhout et al., 2017; D. Lawrence, personal correspondence). We have also added a subsection "Residual" in the Discussion, where we discuss more about the biases that cannot be explained by biases in T or P.

Van Kampenhout, L., J.T.M. Lenaerts, W.H. Lipscomb, W.J. Sacks, D.M. Lawrence, A.G. Slater, and M.R. van den Broeke, 2017. Improving the representation of polar snow and firn in the Community Earth System Model. JAMES, 9, 2583-2600.

Readability: There are also several notations used throughout which can be improved to help the reader. For example, the "model-minus-observations difference" can simply be referred to as model bias. The results section can also be better tied together. Most paragraphs in Section 3 start with "Figure __ shows ...", which becomes very repetitive and causes the paper to lack flow.

We have now used the term "model bias" throughout the paper. We have also revised Results section according to the comment to reduce repetition and make the flow better.

**Minor comments**

L13-14 and throughout: change "SWE change rate in spring" to "spring SWE loss" or similar since the February to May SWE should decrease everywhere.

It is true that SWE decreases everywhere from February to May. However, we are studying the monthly changes and there are regions where SWE can temporarily increase, e.g., from February to March. Therefore, after consideration, we have decided to keep using the term "SWE change rate in spring".

L16: I don't understand what point is being made here: "Even too cold temperatures cannot cause too high SWE without precipitation".

We have removed this statement from the abstract.

L47: State that this is largely because of the increased atmospheric moisture holding capacity.

We have revised the text as suggested.

L48: "Trends in seasonal snow also vary seasonally" awkward wording.

We have revised the text according to the comment as follows:

*Trends in snow cover also vary seasonally*

L48-49: State why spring snow is especially sensitive to warming (e.g., surface albedo feedback is strongest during spring).

We have revised the text according to the comment.

L50: Clarify what is meant by "early-winter"?

We have revised the text according to the comment as follows:

*Early winter from October to December shows even slightly positive trends in both Eurasia and North America, while in January and February, there are no significant trends.*

L70: Change "the difference" to "the model bias"

We have revised the text as suggested throughout the paper.

L72-73: They stated that analysis is needed to understand SWE trends, but this paper only looks at climatological values.

In general, simulated trends of SWE can be considered more reliable if the current climatological distribution of SWE is simulated accurately. We have clarified this in the text.

L88-89: Could be worth showing this for one GCM in the supplement. E.g. a version of Figure 2 where the grey lines represent internal variability rather than intermodal variability.

We have added a version of Fig. 2 in the Supplementary material showing all realizations of three different models (CESM2, MPI-ESM1-2-HR, and EC-Earth3).

[Figure]

**Figure 2. Monthly SWE sum over the entire study area in February, March, April, and May for all realizations of three CMIP6 models.**

115    Table 1: Add model resolution as a column since that is one of the requirements for this study.

We have added a column showing the resolution in the table.

L109: Remove "year"

We have edited the text as suggested.

120

L109-110: Awkward wording, rephrase: "cover non-mountainous regions, and glaciers and ice sheets are excluded."

We have revised the text according to the comment as follows:

*Mountainous regions, glaciers, and ice sheets are excluded from the data.*

125

L119: Sun et al 2018 (doi: 10.1002/2017RG000574) is a good reference for this statement.

We have added this reference in the manuscript.

L120: Why not convert it to mm/month so they are directly comparable?

130    The units mm month$^{-1}$ and kg m$^{-2}$ month$^{-1}$ are equivalent to each other. For precipitation, we have used kg m$^{-2}$ month$^{-1}$ instead of mm month$^{-1}$, because we have used the unit kg m$^{-2}$ for SWE. We have revised the text as follows:

*All precipitation data are presented here in units of kg m$^{-2}$ month$^{-1}$ which is equivalent to mm month$^{-1}$.*

135    L125: Citation needed for this statement.

We have revised the text according to the comment as follows:

*In this study, we have used the monthly mean 2 m air temperature product, which agrees well with other datasets in the arctic (Gelaro et al., 2017; Simmons et al., 2017) and the mean values show very small biases (Bosilovich et al., 2015).*

140

L133: Is there any downside to comparing the models at the observational resolution rather than regridding the observations to match the GCMs?

Resampling data can cause uncertainties in the results regardless of which resolution the data is resampled to. In this case, we decided to use a finer resolution because the resolution of the model is
145    too coarse especially at marginal zone of seasonal snow cover. However, a disadvantage of the resampling approach is that small-scale differences between simulated and observed SWE are interpreted as model biases, even if the grid-mean SWE is simulated correctly. The resampling doesn't however impact the larger-scale biases (i.e., biases at scales resolved by the models). Also, the large amount of data used in this study decreases uncertainties caused by resampling the data.

150

L138: Is this snow covered area calculated for each GCM or is a common snow covered area used across all models? We know from Mudryk et al. (2020) that snow cover extent is highly variable across CMIP6 models.

155 The snow-covered area is calculated individually for each model. We have clarified this in the text.

L144: Shouldn't February be included in this as well since you are assessing the February mean rather than Feb 1 SWE?

We could include February as well, however, as we are using the mean SWE of the whole month and 160 not Feb 28 SWE, there would still be a slight mismatch in the variables.

L159 and throughout: change "model-minus-observation difference" to "model bias".

We have revised the text according to the comment.

165 L188: The precipitation and temperature biases seem fairly important to the overall story so it might be worth promoting this material to the main text.

Figure 1 in the submitted manuscript is meant to only show as an example the spatial distribution of SWE in CMIP6 models and in the SWE reference data. We have considered adding P and T also in Fig. 1 but decided to show them only in the Supplementary material to keep the number of figures more 170 reasonable. The P and T biases for each model separately for winter and spring are shown in Figs. 4, 5, 9 and 10 in the manuscript.

Fig 3: "Mean difference in SWE" should be referred to as "SWE Bias" throughout

We have revised the text according to the comment.

175

Fig 3-4: Slightly confusing how "SWE in winter" refers to February, but "Mean P in winter" refers to the Nov-Jan mean.

We have replaced "Mean P in winter" with "Mean P in Nov-Jan".

180 L221-222: Can you quantify this bias in terms of a percent of the climatology?

We calculated the bias in terms of a percent of the climatology for these models in the northern regions and revised the text as follows:

*For both models, the bias is very high in large regions in northern parts of North America and Eurasia. In these areas, the relative bias is typically 150- 200%.*

185

L225: "Overall, the GFDL models are the most consistent with the SnowCCI data" – add "during February" after this statement.

We have revised the text according to the comment.

190    L230: add "NH extratropical" between overestimate and precipitation.

We have revised the text according to the comment.

L231: remove "dotted"

We have revised the text according to the comment.

195

L237: reword "either too high SWE and too low T or too low SWE and too high T"

We have revised the text according to the comment as follows:

*either cold bias and positive SWE bias or warm bias and negative SWE bias, i.e., the areas where T bias could logically explain the SWE bias.*

200

L251-252: "whereas in other models, deltaSWE is clearly smaller." This is not the most meaningful insight, can you be more detailed.

We have revised the text according to the comment as follows:

*The mean $\Delta SWE$ varies from under 30 kg m$^{-2}$ in the GFDL-CM4 model to around 50 kg m$^{-2}$ in the*
205    *CESM2 and NorESM2-MM models.*

L277: Is it realistic to treat T and P as independent variables?

T and P are not fully independent but are linked to each other through several mechanisms. Especially in large scale and in long time periods T and P are linked to each other, but in shorter time periods and
210    small scale (a grid cell), the dependency between these variables is complicated and varies spatially and seasonally. We have now revised this part to bring up this limitation:

*In fact, this exposes a limitation of the regression Eq. (1): it treats $\Delta T_{cum}$ and $\Delta P_{cum}$ as independent variables, which is not fully realistic. When these variables are correlated, their contributions to $\Delta SWE$ cannot be fully separated.*

215

L280-284: Hypothesize what is unique about these models that could be driving this.

We have added a subsection "Residual" in the Discussion, where we discuss about the possible factors behind these large positive SWE biases.

220    Prior to Figure 8: it seems like there should be a figure showing spring SWE change from OBS and models before showing the biases.

We have added a figure to the Supplementary material showing the SWE$_{change}$ in models and in SWE reference data.

225

L294: DeltaSWEchange is confusing notation. Consider alternatives such as DeltaSWEmelt?

We considered this but decided to keep using DeltaSWEchange.

L298-299 and elsewhere: change "melts more slowly" to "there is less snowmelt". What is shown does not necessarily mean snow is melting faster because they all have different SWEmax values.

We have revised the text as suggested.

L316: change "mutually biases" to "mutual biases"

We have revised the text as suggested.

L327-348: Discussion of EC-Earth biases could mention that these models drastically overestimate NH snow cover extent.

We have revised the text as suggested. We have also added a subsection "Residual" in the Discussion, where we discuss more about the possible factors behind the EC-Earth biases.

L337: "biases in snow melt rate in spring are dominated by other factors than T or P" – further discuss some possible factors in the text (e.g. snow-covered surface albedo biases, which have been documented by numerous studies, albedo feedback strength).

We have added more discussion about the possible factors in the new subsection "Residual". Parameterization of surface albedo and the representation of the albedo feedback are among these factors. This is discussed based on the papers by Thackeray et al. (2015, 2018, 2021).

---

## Author Comment (AC2)

The review comments are shown in black, the author responses in blue and text from revised manuscript in *blue italics*.

We would like to thank the reviewer for reviewing our manuscript. We appreciate all the comments and have revised our manuscript according to them. Please find below our responses to the comments.

**Referee #2**

**Major comments**

**Introduction**

The current section has extremely limited information about the previous studies for climate model-
driven snow products in the Introduction section (such as the general performance of SWE products from earth system models within the CMIP, and what are the previous findings of the differences in CMIP6 as compared to CMIP5 snow products, etc). I would strongly recommend including a further description about climate modeldriven snow products and comparison studies (CMIP5 & 6, and statistical or physically downscaled products e.g. CORDEX) with its reliability and uncertainties in
Introduction section. Also, the authors should provide much more sufficient information about a recent progress of the SnowCCI products from Luojus et al., (2021) and Pulliainen et al. (2020) [this manuscript should provide that information as a standalone work]. I'm sure this will draw potential readers' attention to the necessity of this study.

We have revised the Introduction section according to the comment. We have, for example, included more information about model/reanalysis snow products and, also, added comparison studies between CMIP5 and CMIP6 models.

**Non-mountainous regions**

The authors clearly stated that a main differentiation of the current study from one previous study comparing SWE in CMIP6 models (Mudryk et al., 2020) is to consider both temperature and precipitation to explain the differences in SWE. However, I would note that, unlike Mudryk et al. (2020), this study was conducted only for non-mountainous regions because of the unavailability of the SnowCCI SWE product over complex topography. This is crucial for SWE because a large portion of
the seasonal snow exists mountains (for example, 40 to 60% in North America; Wrzesien et al., 2018; Kim et al., 2021). To achieve the comprehensive results across the NH, thus, I strongly suggest that the authors would consider adapting the weight-based blending approach used in Mudryk et al. (2020) with one or more additional reliable SWE products to include mountainous regions in this study. They used this approach to overcome the unavailability of the Globsnow SWE in complex terrains. The approach allowed them to merge multiple observations and reanalysis products to be able to evaluate CMIP6

SWE over the entire NH domain (not just non-mountainous areas). As the authors may know, the method is that a weight given to the GlobSnow data is linearly reduced with increasing the fraction of mountainous terrain, reaching zero for grid cells containing only mountainous terrain. Regarding dominant portions of the seasonal snow in NH exist in mountain regions, this will surely strengthen the results. Otherwise, it should be clearly stated that this study focuses on non-mountainous regions.

- Wrzesien, M. L., Durand, M. T., Pavelsky, T. M., Kapnick, S. B., Zhang, Y., Guo, J., and Shum, C. K.: A new estimate of North American mountain snow accumulation from regional climate model simulations, Geophys. Res. Lett., 45, 1423–1432, 2018.

- Kim, R. S., Kumar, S., Vuyovich, C., Houser, P., Lundquist, J., Mudryk, L., Durand, M., Barros, A., Kim, E. J., Forman, B. A., Gutmann, E. D., Wrzesien, M. L., Garnaud, C., Sandells, M., Marshall, H.-P., Cristea, N., Pflug, J. M., Johnston, J., Cao, Y., Mocko, D., and Wang, S.: Snow Ensemble Uncertainty Project (SEUP): quantification of snow water equivalent uncertainty across North America via ensemble land surface modeling, The Cryosphere, 15, 771–791, https://doi.org/10.5194/tc-15-771-2021, 2021.

- Mudryk, L., Santolaria-Otín, M., Krinner, G., Ménégoz, M., Derksen, C., Brutel-Vuilmet, C., ... & Essery, R. (2020). Historical Northern Hemisphere snow cover trends and projected changes in the CMIP6 multi-model ensemble. *The Cryosphere*, *14*(7), 2495-2514.

We have added the mountainous regions to the analysis to strengthen our results. As the SnowCCI data are only available over non-mountainous areas, we used MERRA-2, Brown and Crocus v7 datasets to fill the mountainous grid cells. These same datasets were used in Pulliainen et al. (2020). We calculated the mean SWE of these three datasets for each grid cell that were defined as mountainous area in

SnowCCI. Thus, we have now included also the mountainous regions in the analysis, but the results and the conclusions remained quite similar.

**Forested areas**

I am not fully sure about the reliability of the SnowCCI product is enough as a single reference dataset to evaluate the CMIP6 SWE product to achieve a general conclusion, particularly in not only mountainous areas (which were already masked), but also vegetated (or forested) areas in this study.

There are well-known limitations of satellitebased passive microwave (PMW) sensors for snow remote sensing which have been used to develop the GlobSnow product as the main component. Numerous previous studies have found that the passive microwave SWE products are problematic due to many issues (e.g. deep snow "saturation effect", wet snow, forest canopy, terrain heterogeneity, etc.; Dong et al., 2005; Derksen et al., 2010). I believe many readers may also concern about the issues regarding the reliability of the SnowCCI product, particularly in snow hydrology community (Larue et al., 2017).

To address the issue of the product in forested areas, ideally, employing a model/reanalysis SWE product could mitigate it (such as MERRA2 or ERA5; Colleen et al., 2019). Also, it might be helpful to discuss about recent findings in the Introduction or Discussion sections. For example, a recent study from an independent group found that there were better performances of the GlobSnow SWE product as compared to the passive microwave alone SWE retrievals, particularly in maritime and warm forest environments (Cho et al., 2020; this study used the previous version; GlobSnow v2). I strongly recommend providing clear descriptions how (not) to deal with the issues with sufficient literatures.

Dong, J.P. Walker, P.R. Houser, Factors affecting remotely sensed snow water equivalent uncertainty, Remote Sens. Environ., 97 (1) (2005), pp. 68-82

Derksen, P. Toose, A. Rees, L. Wang, M. English, A. Walker, M. Sturm Development of a tundra-specific snow water equivalent retrieval algorithm for satellite passive microwave data, Remote Sens. Environ., 114 (8) (2010), pp. 1699-1709

Larue, F., Royer, A., De Sève, D., Langlois, A., Roy, A., & Brucker, L. (2017). Validation of GlobSnow-2 snow water equivalent over Eastern Canada. *Remote sensing of environment*, *194*, 264-

277.

Cho, E., Jacobs, J. M., & Vuyovich, C. M. (2020). The value of long€ term (40 years) airborne gamma radiation SWE record for evaluating three observation€ based gridded SWE data sets by seasonal snow and land cover classifications. Water resources research, 56(1).

It is true that satellite-based SWE estimates have had problems with several issues. However, all the studies mentioned here are conducted using GlobSnow v2 product, which is not bias-corrected. The bias-correction method has been found to clearly improve the SWE estimation and to solve many issues, which were previously associated with satellite-based SWE estimates. The bias-correction has removed the deep snow saturation effect that was previously an issue for the satellite-based SWE estimates.

Figure 1a in the Extended Data in Pulliainen et al. (2020) shows that the bias-correction method clearly improves the SWE estimates when SWE $>150$ kg m$^{-2}$. We have also added fractional forest cover to the analysis and studied the effect of fractional forest cover on the residual term (Fig. 1).

**Reorganization of the structure of the manuscript**

I think the current manuscript should be re-organized. There exist many statements in Discussion section which are supposed to be in "Result" section (or already mentioned here). There is a limited discussion in the current manuscript which should have been here such as "comparison to previous findings and why they are similar/different", "Limitations in the methods and results", and "future perspectives". To make a more structured manuscript, I would recommend separating Data and Method and making subsections within "Data" section such as "SnowCCI", "MERRA-2 temperature", "GPCC precipitation", and "CMIP6". Also for "Discussion" section, I suggest separating the current form into subsections based on the major findings such as "CMIP6 performance", "Relative contribution of P and T to SWE", and "Limitations and future perspectives", something like them. This would help readers explicitly find and understand this work.

We have now reorganized the manuscript and added subsections according to the comment to make it more clear and easier for readers to understand our work.

**The residual term**

There are many parts that just speculated the reasons of the residual term without supporting explanation based on previous findings or sensitivity analysis (e.g. L254-255, L413-414), even though the portion of the term was considerable. (1) Please provide reasonable rationales to support the author's statements. Regarding this, I think land characteristics such as forest fraction and/or spatial heterogeneity also can impact on generating the residual. To examine this, (2) I would suggest that the authors conduct some sensitivity analysis to provide useful information to be able to explain regional differences in residual from Figures 7 and 12.

We have revised our manuscript according to the comment. We have added a new subsection "Residual" under the Discussion section, where we discuss more the residual term. We have also added forest cover data to the analysis and studied the effect of fractional forest cover on the residual term. Figure 3 shows the dependency between fractional forest cover and the residual for winter and spring. We have calculated the dependency between residual term and fractional forest cover for the entire study area (top row in Fig. 1) and for the non-mountainous area (bottom row). We will add this figure also in the manuscript.

While some correlation between forest cover and residual seems apparent in some models such as CESM2 and NorESM2 (in winter), inter-model differences are still the primary cause of the large spread in the residual term (Fig. 1). The spring-period residual correlations with forest cover, if any, could conceivably result from, for example, snow surface albedo treatment differences in the models. The treatment of snow surface albedo should manifest strongest over open tundra regions and less so over dense forest cover, as we now note elsewhere and have added to the discussion section. However, we emphasize that more detailed per-model investigations on this topic for all participating CMIP6 models in this study are not feasible within the present scope and purpose of the study.

[Figure]

**Figure 1. The dependency between the fractional forest cover and the residual term. Here, we show only one model from each modeling group to keep the number of the models more reasonable.**

**Specific comments**

L13 Specify in-situ "snow depth"

We have edited the text as suggested.

L54 Even though a satellite remote sensing technique is the only option for "observing" SWE at continental scale, state-of-the-art model/reanalysis SWE products have been successfully estimated, and they have been widely used for hydrological and climate research rather than satellite-based approach (mostly passive microwave) probably due to its limitations above. I would suggest rewriting this part covering not only remote sensing approach but also model/reanalysis products for NH SWE.
- Huning, L. S., & AghaKouchak, A. (2020). Global snow drought hot spots and characteristics. Proceedings of the National Academy of Sciences, 117(33), 19753-19759.

We have revised the text according to the comment as follows:

*Observing SWE at continental scale is only possible from satellites, but also model and reanalysis SWE products provide gridded SWE estimates and have been widely used in hydrological and climate research (e.g. Huning and AghaKouchak, 2020; Mortimer et al., 2020; Mudryk et al., 2020). Previously, substantial uncertainties have been reported in NH SWE estimates (Bormann et al., 2018;*

*Mudryk et al., 2015). However, our knowledge of the NH SWE has recently improved considerably, with new bias corrections which reduce the uncertainty of the SWE estimate integrated over NH from 33% to 7.4% (Pulliainen et al., 2020). The bias-correction method, for example, considerably improves SWE estimates in the moderate and deep SWE range (Pulliainen et al., 2020), which has previously caused low bias in SWE estimates (Cho et al., 2020). However, limitations still exist: the bias-correction*

*method cannot be applied in mountainous regions due to the lack of snow course measurements and the large SWE variability in complex terrain (Pulliainen et al., 2020). Even though the area of mountainous regions is limited, these regions store a considerable portion of the seasonal snow (Kim et al., 2021). The bias-correction method mostly increases SWE, and it is therefore likely that without the bias-correction, SWE in mountainous areas is biased low (Pulliainen et al., 2020; Wrzesien et al.,*

*2018).*

L69 They have used four model/reanalysis and satellite SWE datasets and combined them using a blend approach, not just satellite-based data.

We have revised the text according to the comment.

L87 – 89 I think presenting the results from the brief analysis (even in supplementary info) should be helpful for keen reader. Also please provide the detailed description of how the difference among the ensemble members are quantitively smaller than that of models.

We have added a figure (Fig. 2) to the Supplementary material showing all realizations of three different models (CESM2, MPI-ESM1-2-HR, and EC-Earth3). The figure shows that internal variability of each model is smaller than the intermodel variability.

[Figure]

[Figure]

**Figure 2. Monthly SWE sum over the entire study area in February, March, April, and May for all realizations of three CMIP6 models.**

L91-92 Is the GlobSnow v3.0 the same product as SnowCCI used in this study? If not, please add the differences.

Yes, it is the same product. This is mentioned in the revised manuscript.

L100 – 102 Even though the GlobSnow retrieval was improved by combining in-situ snow depth observations as compared to a satellite-only retrieval SWE, there was still large uncertainties for moderate and deep SWE range (about > 150 mm) which is probably due to the "saturation effect" of the volume scattering approach (Derksen et al., 2010; Cho et al., 2020). Was the SnowCCI improve these limitations as compared to the previous version of the GlobSnow? Based on the SWE assessment in Luojus et al. (2021), the overall RMSE for all samples and for shallow to moderate snow conditions only (SWE below 150 mm) is 52.6 mm and 32.7 mm, respectively.

The bias-correction method improves the SWE estimates significantly when SWE > 150 mm. Please see Fig. 1a of Extended Data in Pulliainen et al. (2020).

L109-110 What percentage of the seasonal snow-covered area is non-mountainous area over NH? It would be helpful for reader to get the conclusion from this study within nonmountainous areas (if the authors adhere to non-mountainous area).

We have added the mountainous regions to the analysis.

L112-113, L361-362 Overall, I felt that the paper is overvaluing the accuracy of the SnowCCI product as reference dataset. Please tone down.

We have added three other datasets to cover the mountainous areas, and added also more discussion
about the uncertainties of the datasets in the Introduction and Discussion sections.

L254 What does "model structural factors" mean? Be specific.

With model structural factors we refer to modelling deficiencies that cause errors in the simulated SWE, other than biased simulation of T or P.  In particular, these factors may include deficiencies in the
parameterization of surface energy budget and other snow-related physical processes (phase of precipitation, snow cover fraction, snow albedo, heat conduction in snow, etc.). This is discussed in more depth in the subsection "Residual term" of the revised manuscript. We have edited this sentence as follows:

*The large residual term implies that observational uncertainty or model structural factors, such as*
*deficiencies in the parameterization of surface energy budget and other snow-related physical processes, play a considerable part in the observed SWE biases.*

L254-255 This is speculation for me. Please provide rationale based on literatures related to this statement.

In our understanding, this statement is not speculation - in fact, there aren't really other possibilities. The results clearly indicate that the residual term is substantial and, also, that it varies between the models. Thus, even if the models were able to simulate temperature and precipitation correctly, the simulated SWE would differ from observations. Obviously, any errors in the observations would impact this residual. However, since the residual varies substantially between the models, it is also clear that
the residual is influenced by modelling inaccuracy (other than errors in temperature and precipitation).

L259 I do not think $R^2$ is a "parameter" of linear regression.

We have edited the text according to the comment.

Figure 11 To me, the residual terms overwhelmed the contribution of P and T. In this case, are the contributions of P and T still statistically significant?

It is true that the residual term is considerably larger than the contributions of T and P. Also, $R^2$ values are quite low in spring, suggesting that bias in SWE change rate does not depend much on bias in T or P. We have added figures showing the statistical significance of the terms in the Supplementary material.

L337 Please add further discussion "other factors" particularly in spring season. Do you think mismatching of the spatial resolution among the data sets can be one of the reasons? If so, please add some discussion about this. Regarding this, how do you think of the resampling method (nearest neighbor)?

We have added more discussion about the residual term in the Discussion section. The resampling can influence the small-scale (i.e., model subgrid-scale) features of the residual, but not the larger-scale features. Since the residual clearly shows large-scale structures in many cases, the resampling does not appear to be a major factor explaining the residual.

Figure S6 There are areas where the R^2 values are extremely low. I think it would be good to show the beta_P and beta_T for regions only where there are statistically significant. Please consider applying this throughout all figures.

We have added figures showing $\beta_P$ and $\beta_T$ only for regions, where there are statistically significant, in the Supplementary material.

L342 Be consistent either "Fig" or "Figure"

We have edited the text according to the comment and used "Fig" throughout the text.

L360-361 This sentence is redundant as the authors already mentioned. I would suggest rephrasing something like "while …, our study focuses on analyzing the CMIP6 SWE responses to both temperature and precipitation"

We have revised the text according to the comment.

L362-364 I am not sure if the statements are needed here, which were already mentioned several times.

We have removed this statement from the text.

L373 Figs.

We have revised the text according to the comment.

L360-364 & 376-380 To me, it seems like the summary, not discussion. I would strongly recommend using here for the detailed discussion, such as what are similar/different and what are new findings from this study as compared to previous studies?

We have edited Discussion according to the comment. We have, for example, added more discussion about the residual term, the uncertainties of the reference datasets and compared our results more with previous studies.

L388 Figs. If you refer more than two figures, please use Fig"s"

We have edited the text according to the comment.

L430 I suggest providing much more details of the limitations/uncertainties from the SnowCCI and others to provide sufficient information for those who would use the data sets for their own research, particularly for the issues that I provided in the major comment (such as uncertainties in forested areas which have been challenging areas in snow community). What would the authors expect potential uncertainties in GPCC? Please add discussion sufficiently.

We have edited Discussion according to the comment and added more details of the limitations and uncertainties from all the datasets used in this study.